# Leveraging Additional Information in POMDPs with Guided Policy Optimization

## Abstract

Reinforcement Learning (RL) in partially observable environments poses significant challenges due to the complexity of learning under uncertainty. While additional information, such as that available in simulations, can enhance training, effectively leveraging it remains an open problem. To address this, we introduce Guided Policy Optimization (GPO), a framework that co-trains a guider and a learner. The guider takes advantage of supplementary information while ensuring alignment with the learner's policy, which is primarily trained via Imitation Learning (IL). We theoretically demonstrate that this learning scheme achieves optimality comparable to direct RL, thereby overcoming key limitations inherent in IL approaches. Our approach includes two practical variants, GPO-penalty and GPO-clip, and empirical evaluations show strong performance across various tasks, including continuous control with partial observability and noise, and memory-based challenges, significantly outperforming existing methods.

## 1 Introduction

Many real-world tasks can be formulated as sequential decision-making problems where agents must take actions in an environment to achieve specific goals over time (Puterman, 2014). Reinforcement Learning (RL) has emerged as a powerful tool for solving such tasks, leveraging trial-and-error learning to optimize long-term rewards (Sutton & Barto, 2018). Despite its success, RL encounters significant hurdles in complex and partially observable environments, where agents often operate with limited or noisy information (Madani et al., 1999). However, during training, we often have access to supplementary information that could significantly enhance learning efficiency and performance (Lee et al., 2020; Chen et al., 2022). For instance, in robotics, while real-world sensor data may be noisy or incomplete, simulation environments typically provide full state observability.

Although this extra information offers the potential to accelerate learning, effectively leveraging it in practice remains a major challenge. By introducing a teacher with access to additional information, Imitation Learning (IL) (Hussein et al., 2017) offers a promising approach to address this challenge, as it is often more sample-efficient than traditional RL by enabling agents to learn directly from a teacher's actions. Yet, this approach presents new difficulties: a suboptimal teacher may propagate flawed strategies (Rajeswaran et al., 2017), while a teacher with extra information may set an unrealistically high standard, making it difficult for the agent to imitate effectively. The latter issue, known as an "impossibly good" teacher (Walsman et al., 2023) or imitation gap (Weihs et al., 2024), can impede learning and degrade performance. Prior efforts to address these issues have integrated RL with IL (Weihs et al., 2024; Shenfeld et al., 2023a; Nguyen et al., 2023), but typically assume access to a pre-trained teacher, which may not always be feasible. While one could train a teacher using additional information before training the agent, this two-step process is often inefficient and computationally expensive.

To better utilize available information, we consider a more integrated approach: training a "possibly good" teacher that the agent can consistently follow. Drawing inspiration from Guided Policy Search (GPS) (Levine & Koltun, 2013; Montgomery & Levine, 2016), we introduce Guided Policy Optimization (GPO), a framework that alternates between RL for the teacher and IL for the agent, ensuring the teacher remains aligned with the agent's policy. The key insight is that by leveraging the additional information during training, the teacher can be more easily trained, while maintaining a level of performance that is "possibly good" rather than perfect, being more straightforward for

the agent to follow. Theoretically, we show that the agent can achieve optimality akin to direct RL training, thus mitigating suboptimality and imitation gaps often faced by purely supervised agents. Building on this framework, we present a practical implementation of GPO using Proximal Policy Optimization (PPO) (Schulman et al., 2017), with two variants: GPO-penalty and GPO-clip. These methods introduce minimal modifications, making them efficient and straightforward to apply.

We empirically validate our algorithm across various tasks. In tasks where traditional guidance methods fail to produce optimal policies, our approach proves highly effective. We further validate our algorithm on challenging continuous control tasks in partially observable, noisy environments within the MuJoCo (Todorov et al., 2012) domain, where GPO outperforms baseline methods. Additionally, in memory-based tasks from the POPGym (Morad et al., 2023) benchmark, GPO shows significant improvements, underscoring its ability to exploit extra information and deliver robust performance across diverse domains.

## 2 BACKGROUND

We consider Partially Observable Markov Decision Process (POMDP) (Kaelbling et al., 1998), which is characterized by the tuple $\langle \mathcal{S}, \mathcal{A}, r, \mathcal{P}, \mathcal{O}, \gamma \rangle$. $\mathcal{S}$ represents the set of states, $\mathcal{A}$ the set of actions, $r$ the reward function, $\mathcal{P}$ the transition probability function, $\mathcal{O}$ the partial observation function and $\gamma$ the discount factor. At each time step $t$, agent receives a partial observation $o_t \sim \mathcal{O}(\cdot|s_t)$ for current state $s_t \in \mathcal{S}$. The agent then selects an action $a_t \in \mathcal{A}$ according to $o_t$ or its action-observation history $\tau_t : \{o_0, a_0, o_1, a_1 ..., o_t\}$. The state transitions to the next state $s_{t+1}$ according to $\mathcal{P}(s_{t+1}|s_t, a_t)$, and agent receives a reward $r_t$. The goal for the agent is to find the optimal policy $\pi^* : \tau \to \Delta(\mathcal{A})$ that maximizes the policy value, expressed as $\pi^* = \arg\max_\pi V_\pi$, where $V_\pi = \mathbb{E}[\sum_{t=0}^\infty \gamma^t r_t|\pi]$ represents cumulative rewards.

Assuming that the state $s$ is available during training, we can train a policy $\mu : s \to \Delta(\mathcal{A})$ based on this state information. For clarity, we denote $\mu$ as the **guider** and $\pi$ as the **learner** throughout the paper. Unlike IL, we do not assume access to any additional policy; thus, the guider must be trained from scratch. For convenience, while the observations available to the guider could be any form of privileged information, we directly refer to the state $s$ in the remainder of this paper.

### 2.1 IMITATION LEARNING

Imitation learning (IL) (Hussein et al., 2017) requires having either an expert policy that can effectively accomplish a task or example trajectories produced by this expert policy. A straightforward approach to training the agent is to directly supervise the agent policy $\pi$ using expert policy $\mu$, similar to Behavioral Cloning (BC) (Pomerleau, 1991; Torabi et al., 2018):

$$\min_\pi \mathbb{E}_{s \sim d_\mu}[D_{\mathrm{KL}}(\mu(\cdot|s), \pi(\cdot|s))] = \min_\pi \mathbb{E}_{s \sim d_\mu, a \sim \mu}\left[\log\left(\frac{\mu(a|s)}{\pi(a|s)}\right)\right]. \tag{1}$$

This formulation can also be interpreted as a maximum likelihood estimation problem in supervised learning. The $d_\mu(s) := (1-\gamma)\sum_t \gamma^t \mathrm{Pr}(s_t = s; \mu)$ is discounted stationary state distribution induced by the expert policy $\mu$. However, if the expert has access to privileged information that the agent lacks, the agent can only learn the statistical average of the expert's actions for each observable state $o$. Specifically, this leads to $\pi(\cdot|o) = \mathbb{E}_{d_\mu}[\mu(\cdot|s)|o = f(s)]$, where $f(s)$ denotes the observable function of the state (Warrington et al., 2020; Weihs et al., 2024). This limitation may result in sub-optimal performance, which we will illustrate with two examples in the following section.

### 2.2 DIDACTIC EXAMPLES

**TigerDoor**. In the classic TigerDoor problem (Littman et al., 1995), there are two doors with a tiger hidden behind one of them. The possible state $s_L$ (tiger behind the left door) and $s_R$ (tiger behind the right door), with equal probabilities for each, forming $\mathcal{S} = \{s_L, s_R\}$. The action set is $\mathcal{A} = \{a_L, a_R, a_l\}$, where $a_L$ and $a_R$ denote opening the left and right doors, respectively, and $a_l$ denotes listening to determine the tiger's location. The guider knows the tiger's location whereas the learner can only ascertain it after choosing $a_l$. The payoff matrix is shown in Table 1. The optimal policy for the guider is to always choose the correct door without listening, whereas the learner's optimal strategy involves first listening to locate the tiger. Consequently, the learner cannot learn

the optimal policy through supervision from the guider, as the guider never chooses $a_l$. Under the guider's supervision, the learner will only learn to randomly select between $a_L$ and $a_R$, resulting in an expected reward of 0.5. This scenario poses challenges for the supervised learner, as the guider fails to explore and gather essential information for the learner.

| state \ action | $a_L$ | $a_R$ | $a_l$ |
|---|---|---|---|
| $s_L$ | 1 | 0 | -0.1 |
| $s_R$ | 0 | 1 | -0.1 |

Table 1: TigerDoor problem

| state \ action | $a_L$ | $a_R$ |
|---|---|---|
| $s_L$ | 2 | 0 |
| $s_R$ | 0 | 1 |

Table 2: TigerDoor-alt problem

**TigerDoor-alt**. We introduce an alternative version of the problem, called TigerDoor-alt, which also highlights an imitation gap, even without additional exploratory information. In this scenario, the listening action $a_l$ is removed, and the reward for correctly selecting the left door is increased to 2 as shown in Table 2. Similarly, the guider continues to select the correct door, while the learner learns to randomly choose between the two doors, yielding an expected reward of 0.75. However, the optimal policy for the learner is to always choose the left door, which provides an expected reward of 1. This discrepancy arises from the loss of information when converting the reward-based objective into a policy-supervised objective.

While these issues can be addressed by directly applying RL to the learner, as seen in prior work (Weihs et al., 2024; Shenfeld et al., 2023a;b), this approach can negate the efficiency gains of supervised learning, especially in more complex tasks. In Sections 3.1 and 4.1, we will demonstrate that our algorithm can achieve optimality without requiring RL training for the learner, both theoretically and experimentally.

## 3 METHOD

We present our Guided Policy Optimization (GPO) framework, which co-trains two entities: the guider and the learner. Inspired by Guided Policy Search, GPO iteratively updates both policies to ensure alignment. We then explore both the theoretical properties and practical implementation of GPO, introducing two variants: GPO-penalty and GPO-clip.

### 3.1 GUIDED POLICY OPTIMIZATION

The GPO framework operates through an iterative process comprising four key steps:

1. **Data Collection**: Collect trajectories by executing the guider's policy, denoted as $\mu^{(k)}$.

2. **Guider Training**: Update the guider $\mu^{(k)}$ to $\hat{\mu}^{(k)}$ according to RL objective $V_{\mu^{(k)}}$.

3. **Learner Training**: Update the learner to $\pi^{(k+1)}$ by minimizing the distance $D(\pi, \hat{\mu}^{(k)})$.

4. **Guider Backtrack**: Set $\mu^{(k+1)}(\cdot|s) = \pi^{(k+1)}(\cdot|o)$ for all state $s$ before the next iteration.

In step 3, $D(\pi, \mu)$ can be any Bregman divergence. For this work, we utilize the KL divergence weighted by the state distribution $d_\mu$. GPO iterates these steps until convergence, applying standard RL to train the guider, while the learner seeks to mimic the guider's behavior. If the learner struggles due to discrepancies in observation spaces, the backtrack step adjusts the guider's policy to mitigate the imitation gap.

A key feature of GPO is that only the guider's policy interacts with the environment, ensuring that data is always generated from the distribution induced by $\mu$. Importantly, despite the learner not directly interacting with the environment, we demonstrate that GPO achieves the same convergence and optimality guarantees as direct RL training. For simplicity, we assume the guider $\mu$ has access to an unlimited policy class, while the learner $\pi$ is limited to a constrained policy class $\Pi$.

**Proposition 1.** *If the guider's policy is updated using policy mirror descent in each GPO iteration:*

$$\hat{\mu} = arg\min\{-\eta_k\langle\nabla V(\mu^{(k)}), \mu\rangle + \frac{1}{1-\gamma}D_{\mu^{(k)}}(\mu, \mu^{(k)})\},\tag{2}$$

*then the learner's policy update follows a constrained policy mirror descent:*

$$\pi^{(k+1)} = arg\min_{\pi \in \Pi} \{-\eta_k \langle \nabla V(\pi^{(k)}), \pi \rangle + \frac{1}{1-\gamma} D_{\pi^{(k)}}(\pi, \pi^{(k)})\} \tag{3}$$

*Proof.* See Appendix B. □

Policy Mirror Descent (PMD) (Tomar et al., 2020; Xiao, 2022) is a general family of algorithms that covers a wide range of fundamental methods in RL, particularly trust-region algorithms like TRPO (Schulman et al., 2015a) and PPO. This proposition demonstrates that if we use an algorithm belonging to the PMD family for updating the guider's policy, the iterative process of GPO can be viewed as applying the same algorithm directly to the learner. In other words, the update of the learner's policy can inherit the properties such as monotonic policy improvement (Schulman et al., 2015a) from trust-region algorithms. This suggests that GPO can effectively address challenges in IL, such as dealing with a suboptimal teacher or the imitation gap, while still framing the learner's policy as being supervised by the guider. In Appendix **??**, we provide an intuitive example to show how GPO can achieve optimal in TigherDoor-alt problem.

Given that GPO effectively mirrors direct RL for the learner, one may ask: **What are the key advantages of GPO?** The primary benefit is that GPO simplifies the learning process by leveraging additional information. The guider's training is generally easier than the learner's, particularly since policy gradients for the learner suffer from high variance, worsened by partial observability. By dividing the learning process, GPO handles this challenge more effectively. The guider is updated using policy gradients, while the learner is trained through supervised learning, thereby assigning more complex tasks to the guider and simplifying the learner's objective. For example, when training an agent to be robust to noise, we may deliberately add noise to the observations. However, this will complicate training due to the noise in both observations and policy gradients. GPO addresses this by training the guider without noise and supervising the learner with noisy observations, making the process more manageable and robust.

### 3.2 IMPLEMENTATION OF GPO

This section discusses the implementation of the GPO framework. In step 2 of GPO, we use PPO as the underlying trust-region algorithm. The corresponding objective for the guider's policy is as follows[1]:

$$L_1(\mu) = \mathbb{E}\left[ \min\left( r^\mu(s,a) A^\beta(s,a), r^\mu_{clip}(s,a,\epsilon) A^\beta(s,a) \right) \right], \tag{4}$$

where $r^\mu(s,a) = \mu(a|s)/\beta(a|s)$, $r^\mu_{clip}(s,a,\epsilon) = clip(r^\mu(s,a), 1-\epsilon, 1+\epsilon)$ and $\beta$ denotes the behavioral policy. The advantage $A^\beta(s,a)$ is estimated using the Generalized Advantage Estimation (GAE) (Schulman et al., 2015b) with the value function $V(s)$ trained via discounted reward-to-go.

In step 3, since finding the exact minimizer of the distance measure is computationally prohibitive, we use gradient descent to minimize the BC objective: $L_2(\pi) = \mathbb{E}\left[ D_{KL}\left( \mu(\cdot|s), \pi(\cdot|o) \right) \right]$. Similarly, in step 4, we backtrack the guider's policy using the same BC loss: $L_3(\mu) = \mathbb{E}\left[ D_{KL}\left( \mu(\cdot|s), \pi(\cdot|o) \right) \right]$.

A key insight in the implementation of GPO is that rigorous backtracking of the guider's policy is unnecessary Instead, our goal is to maintain the guider in a "possibly good" region relative to the learner. Two scenarios can explain why the learner may not fully follow the guider: (1) the guider's policy is too optimal for the learner to imitate, or (2) the guider is improving faster than the learner, which is common in practice since gradient descent usually results in inexact minimization. In the second case, excessive backtracking of the guider is counterproductive. Moreover, keeping the guider slightly superior to the learner enables it to collect better trajectories, and we will discuss in Section 4.4. To maintain this balance, we introduce a coefficient $\alpha$ that modulates the guider's objective as $L(\mu) = L_1(\mu) - \alpha L_3(\mu)$, where $\alpha$ is adapted based on the distance $L_3(\mu)$ relative to a threshold $d_{\text{targ}}$, using a constant scaling factor $k$:

$$\alpha = k\alpha \text{ if } L_3(\mu) > k d_{\text{targ}}, \text{ else } \alpha/k \text{ if } L_3(\mu) < d_{\text{targ}}/k. \tag{5}$$

---

[1]We omit subscripts for expectations in the remainder of the paper, as all samples are drawn from the distribution induced by the behavioral policy $\beta = \mu_{\text{old}}$.

This scheme is analogous to the KL-penalty adjustment in PPO-penalty (Schulman et al., 2017), where the penalty coefficient adjusts based on the relationship between the KL divergence and a predefined threshold.

Another key aspect is compensating for the learner's policy improvement, as we replace strict backtracking with a KL-constraint. While it is possible to set a very small $d_{\text{targ}}$, this would inefficiently inflate $\alpha$, hindering the guider's training. Notably, Proposition 1 implies that applying GPO with PPO is effectively equivalent to applying PPO directly to the learner. Consequently, we can concurrently train the learner's policy using PPO during the GPO iterations. As a result, we introduce an additional objective for the learner's policy:

$$L_4(\pi) = \mathbb{E}\left[\min\left(r^\pi(s,a)A^\beta(s,a), r^\pi_{clip}(s,a,\epsilon)A^\beta(s,a)\right)\right], \tag{6}$$

where $r^\pi(s,a) = \pi(a|o)/\beta(a|s)$. Considering that the behavioral policy is from guider, to validate this update, we introduce the following proposition:

**Proposition 2.** *For policy $\pi$, $\mu$, $\beta$ and all state $s$, suppose $D_{TV}(\mu(\cdot|s), \beta(\cdot|s)) \lesssim \epsilon/2$, then we have*

$$\mathbb{E}_{a\sim\beta}\left[|1 - r^\pi(s,a)|\right] \lesssim \epsilon + \sqrt{2d_{targ}}. \tag{7}$$

The assumption on total variation distance is justified by the PPO update of the guider's policy (Appendix B). This proposition implies that when $d_{\text{targ}}$ is small, the behavioral policy closely matches the learner's policy, allowing valid sample reuse for learner training.

Finally, we define the merged learner objective for the learner as: $L(\pi) = \alpha L_4(\pi) - L_2(\pi)$, where the coefficient $\alpha$ from (5) is applied to the RL term. This mechanism compensates when the learner struggles to follow the guider. If the learner is able to fully track the guider, $\alpha$ approaches zero, allowing the guider to directly lead the learner to the optimal policy without requiring an additional RL objective. When the learner cannot keep pace, the RL objective aids in the learner's training.

## 3.3 REFINEMENTS OF GPO

In this section, we introduce several refinements to the GPO framework. The key principle guiding these refinements is that an effective guider should remain at the boundary of the learner's "possibly good" region: if the guider is too far ahead, the learner struggles to follow; if too close, the guider's ability to provide effective supervision and better trajectory diminishes. To achieve this balance, the guider should halt updates when it moves too far ahead and avoid backtracking when it is already sufficiently close.

We propose two key modifications to the original algorithm outlined in the previous subsection. First, inspired by PPO-clip, we replace the clip function $r^\mu_{clip}(s,a,\epsilon)$ in (4) with the following double-clip function:

$$r^{\mu,\pi}_{clip}(s,a,\epsilon,\delta) = \text{clip}\left(\text{clip}(\frac{\mu(a|s)}{\pi(a|o)}, 1-\delta, 1+\delta) \cdot \frac{\pi(a|o)}{\beta(a|s)}, 1-\epsilon, 1+\epsilon\right). \tag{8}$$

This formulation introduces an additional inner clipping step, which halts the guider's updates under two conditions: (1) $A^\beta(s,a) > 0$ and $\mu(a|s) > \pi(a|o)(1+\delta)$, (2) $A^\beta(s,a) < 0$ and $\mu(a|s) < \pi(a|o)(1-\delta)$. Considering that positive (negative) advantage indicates that $\mu(a|s)$ is set to increase (decrease), the double-clip function prevents further movement away from $\pi$ when $\mu$ is already distant.

It is important to note that, unlike PPO where PPO-clip can completely replace the KL-penalty term, this is not the case in GPO. In PPO, the ratio $r^\pi(s,a)$ starts at 1 at the beginning of each epoch, ensuring that the clipped ratio keeps $\pi$ near the behavioral policy. In GPO, however, the gap between $\pi(a|s)$ and $\mu(a|o)$ may accumulate over multiple updates if the learner fails to keep up with the guider. The double-clip function (8) alone is insufficient to bring $\pi(a|o)$ back into the $\delta$ region once it has strayed too far. To address this, we introduce a mask on the backtracking loss, defined as: $m(s,a) = \mathbb{I}(\frac{\pi(a|s)}{\mu(a|o)} \notin (1-\delta, 1+\delta))$, where $\mathbb{I}$ is the indicator function. This mask replaces the adaptive coefficient $\alpha$ from the previous subsection, selectively applying the backtracking penalty only when $\mu(a|o)$ drifts outside the $\delta$ region. Policies that remain close to each other are left unaffected, preventing unnecessary backtracking.

Additionally, given that both the guider and learner are solving the same task, their policies should exhibit structural similarities. To leverage this, we allow the guider and learner to share a single policy network. To distinguish between guider and learner inputs, we define a unified input format: the input to the guider's policy is defined as $o_g = [s, o, 1]$, where $s$ is the state, $o$ is the partial observation, and the scalar 1 serves as an indicator; the learner's input is defined as $o_l = [\vec{0}, o, 0]$, where $\vec{0}$ is a zero vector with the same dimensionality as $s$, indicating that the learner has access only to the partial observation $o$.

Finally, we name the method introduced in Section 3.2 as **GPO-penalty**, and the refined method presented here as **GPO-clip**. The update for the shared policy network with parameters $\theta$ is as follows:

$$
\begin{aligned}
L_{\text{GPO-penalty}}(\theta) =\mathbb{E}\Big[ &\min\left(r^{\mu_\theta}A^\beta(o_g,a), r^{\mu_\theta}_{clip}A^\beta(o_g,a)\right) - \alpha D_{\text{KL}}\big(\mu_\theta(\cdot|o_g)||\pi_{\hat{\theta}}(\cdot|o_l)\big) \\
&\alpha \min\left(r^{\pi_\theta}A^\beta(o_l,a), r^{\pi_\theta}_{clip}A^\beta(o_l,a)\right) - D_{\text{KL}}\big(\mu_{\hat{\theta}}(\cdot|o_g)||\pi_\theta(\cdot|o_l)\big)\Big],
\end{aligned}
\tag{9}
$$

$$
\begin{aligned}
L_{\text{GPO-clip}}(\theta) =\mathbb{E}\Big[ &\min\left(r^{\mu_\theta}A^\beta(o_g,a), r^{\mu_\theta,\pi_{\hat{\theta}}}_{clip}A^\beta(o_g,a)\right) - m D_{\text{KL}}\big(\mu_\theta(\cdot|o_g)||\pi_{\hat{\theta}}(\cdot|o_l)\big) \\
&\alpha \min\left(r^{\pi_\theta}A^\beta(o_g,a), r^{\pi_\theta}_{clip}A^\beta(o_g,a)\right) - D_{\text{KL}}\big(\mu_{\hat{\theta}}(\cdot|o_g)||\pi_\theta(\cdot|o_l)\big)\Big],
\end{aligned}
\tag{10}
$$

where $\hat{\theta}$ denotes a stop-gradient operation on the parameters, and $\alpha$ for GPO-clip is a fixed parameter. The complete algorithm is summarized in Appendix C. Generally, converting PPO to GPO requires minimal adjustments—no additional networks or rollouts are necessary, and only a few extra lines of code are needed to compute the additional losses.

## 4 EXPERIMENTS

In this section, we evaluate the empirical performance of our GPO algorithm across various domains. Section 4.1 presents didactic tasks to verify GPO's properties, such as optimality. Section 4.2 evaluates GPO on partially observable and noisy continuous control MuJoCo (Todorov et al., 2012) tasks in the MuJoCo environment, comparing it against several baselines. Section 4.3 evaluates GPO's performance on memory-based tasks from POPGym (Morad et al., 2023), and Section 4.4 provides ablation studies and further discussion.

Given that in our setting, an expert policy is unavailable unless trained from scratch, we consider the following algorithms as baselines. A summary of their main characteristics is presented in Table 3. Among them, **GPO-naive** refers to GPO-penalty without the RL auxiliary loss. **PPO-V** directly trains the learner using PPO, with its value function receiving $o_g$ as input. **PPO+BC** trains the guider with PPO while the learner is trained through direct BC from the guider. **ADVISOR-co** and **A2D** are baselines from previous works Weihs et al. (2024) and Warrington et al. (2020), respectively. Further details about these algorithms are provided in Appendix E.1.

| Algorithm | Train $\mu$ | Behavioral policy | Train $\pi$ | Value function | backtrack $\mu$ |
|-----------|-------------|-------------------|-------------|----------------|-----------------|
| PPO | - | $\pi(a|o_l)$ | PPO | $V(o_l)$ | - |
| PPO+V | - | $\pi(a|o_l)$ | PPO | $V(o_g)$ | - |
| PPO+BC | PPO | $\mu(a|o_g)$ | BC | $V(o_g)$ | No |
| A2D | PPO | $\pi(a|o_l)$ | BC | $V(o_l)$ | No |
| ADVISOR-co | PPO | $\pi(a|o_l)$ | BC+PPO | $V(o_l)$ | No |
| GPO-naive | PPO | $\mu(a|o_g)$ | BC | $V(o_g)$ | Yes |
| GPO-penalty | PPO | $\mu(a|o_g)$ | BC+PPO | $V(o_g)$ | Yes |
| GPO-clip | PPO | $\mu(a|o_g)$ | BC+PPO | $V(o_g)$ | Yes |
| GPO-ablation | PPO | $\mu(a|o_g)$ | PPO | $V(o_g)$ | Yes |

Table 3: Algorithms.

### 4.1 DIDACTIC TASKS

We begin by evaluating our algorithm on two didactic problems introduced in Section 2.2. As shown in Fig. 1(a)(b), direct cloning of the guider's policy converges to a suboptimal solution, as expected.

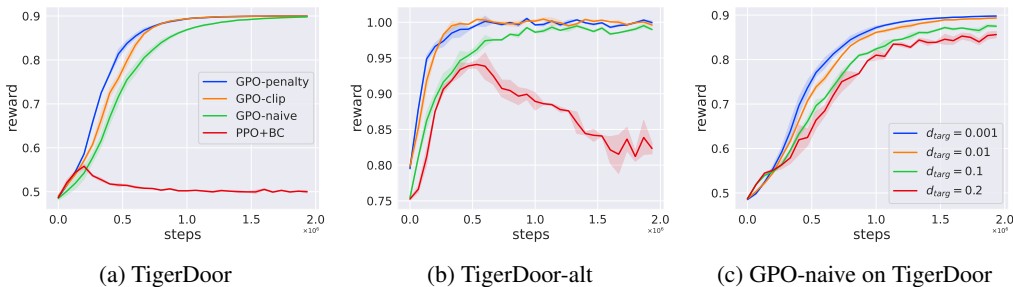

(a) TigerDoor     (b) TigerDoor-alt     (c) GPO-naive on TigerDoor

Figure 1: Results for the TigerDoor and TigerDoor-alt.

In contrast, all variants of GPO achieve optimal performance on these tasks. Although applying RL directly to the learner easily leads to optimal solutions, it is important to note that GPO-naive achieves optimality purely through supervised learning. This result verifies the optimality guarantee of the GPO framework described in Proposition 1, suggesting that a guider constrained within the learner's "possibly good" region can provide effective supervision, even with asymmetric information. Besides, comparing GPO-naive to GPO-penalty and GPO-clip reveals that the introduction of direct RL training for the learner accelerates learning. Moreover, as shown in Fig. 1(c), the optimality of GPO-naive is robust to variations in the KL-threshold, offering flexibility to adjust the distance between the guider and learner across different tasks.

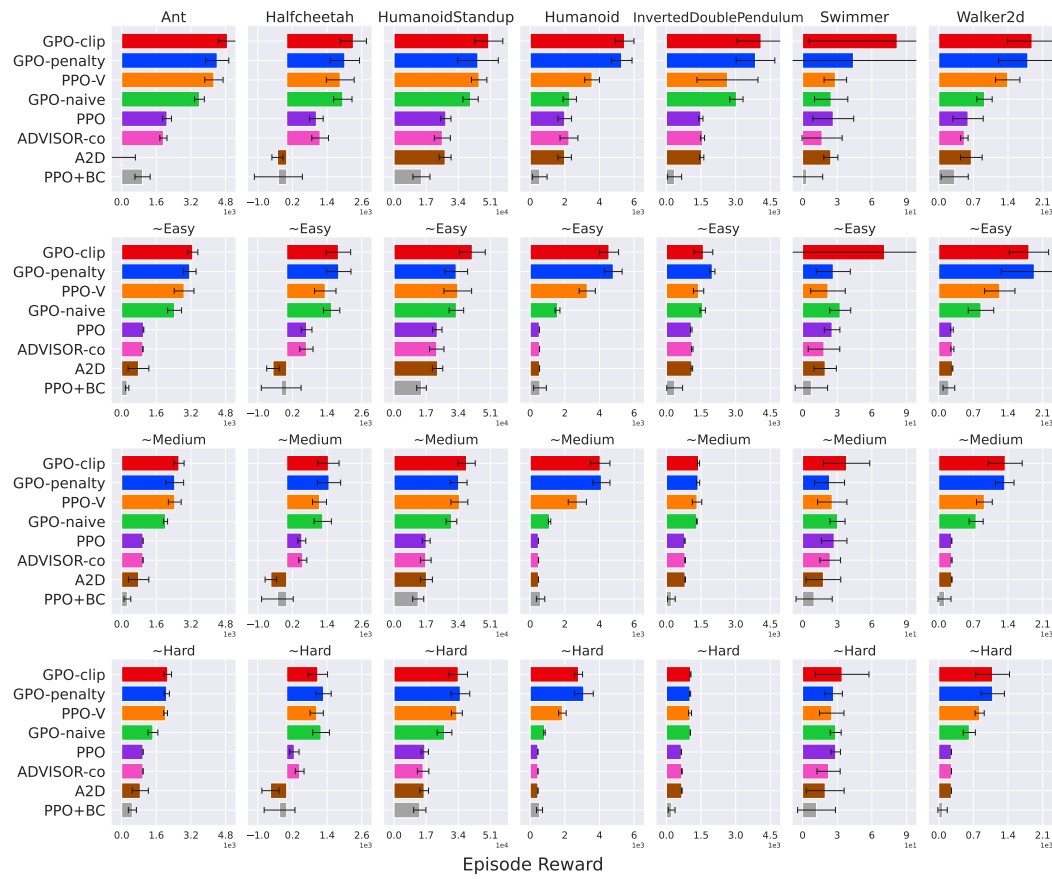

Figure 2: Comparing betweetn GPO and other baselines on 28 MuJoCo tasks. The four figures in each column represent the same task with different levels of noise, where $Easy$, $Medium$ and $Hard$ represent normal noise with standard deviation equals to 0.1, 0.2 and 0.3, respectively.

## 4.2 CONTINUOUS CONTROL TASKS IN MUJOCO

In this subsection, we present the results of our algorithms and baselines on several continuous control tasks in the MuJoCo domain. We adopt the implementation of PPO and MuJoCo environments in Brax (Freeman et al., 2021). To transform the MuJoCo tasks into a POMDP setting, we follow a similar approach to that used in POPGym: the velocity information of all joints is removed, and varying levels of noise are added to the observations. The guider has access to full, noiseless information, while the learner operates with partial and noisy inputs. For more details, please refer to Appendix E.

The results are shown in Fig. 2, where the performance hierarchy is generally GPO-clip > GPO-penalty > PPO-V > GPO-naive > other baselines. We have the following key observations:

**1**. GPO-based methods consistently outperform PPO, demonstrating that GPO effectively utilizes additional information during training, thereby improving the learning efficiency of the agent.

**2**. As a popular approach for utilizing additional information (Pinto et al., 2018; Andrychowicz et al., 2020), especially in MARL (Yu et al., 2022), PPO-V performs relatively well due to its more accurate value function, which reduces the variance in policy gradients thanks to its access to noiseless observations. Since PPO-V is theoretically equivalent to GPO with exact backtracking (Proposition 1), we provide further experiments and discussions in subsequent sections.

**3**. Comparing GPO-naive to GPO-penalty and GPO-clip, we see that introducing RL training for the learner significantly improves performance. It compensates for policy improvement when the guider and learner are not closely aligned, and helps the learner itself make progress when the guider struggles to optimize while staying close to the learner.

**4**. Comparing PPO-BC and GPO-naive highlights the necessity of backtracking. If we train the guider without considering the learner's progress, the imitation gap becomes significant, and the guider's the supervision becomes ineffective, leading to suboptimal performance.

**5**. ADVISOR-co performs similarly to PPO due to the absence of effective backtracking. The guider quickly outpaces the learner, causing the weight coefficient in ADVISOR to diminish, effectively reducing it to pure PPO training.

**6**. A2D performs poorly across all tasks. Although it also allows the co-training of the guider and learner, it fails to maintain a good guider policy. Since its behavioral policy comes from the learner, without proper backtracking, the guider soon drifts outside the "possibly good" region, rendering its RL training ineffective as the behavioral policy diverges from the current update policy.

In summary, our method consistently outperforms the baselines across almost all tasks, highlighting its effectiveness in solving noisy and partially observable continuous control tasks.

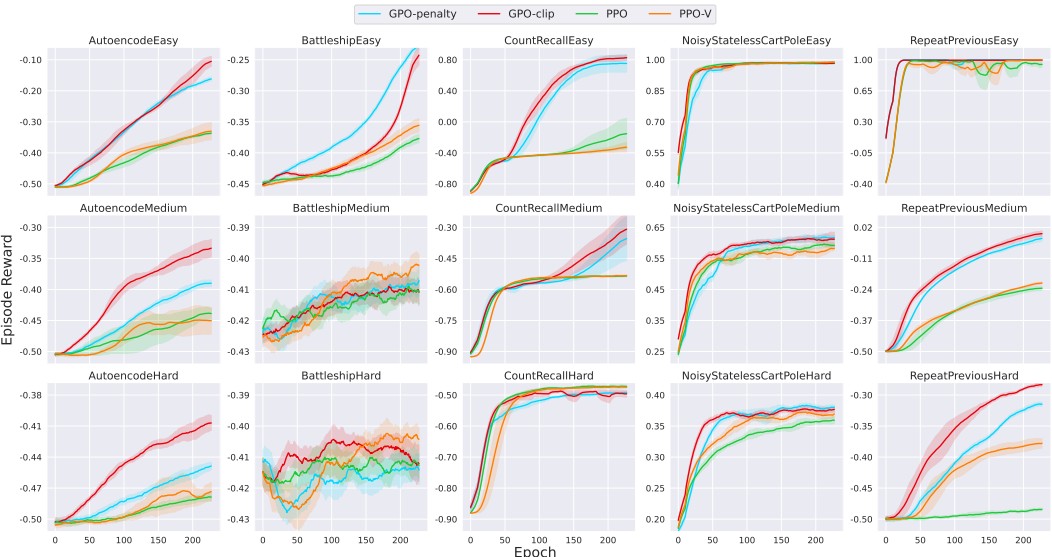

Figure 3: The results of GPO-penalty, GPO-clip, PPO-V and PPO on 15 POPGym tasks.

### 4.3 MEMORY-BASED TASKS IN POPGYM

In this subsection, we evaluate GPO on several memory-based tasks from POPGym, using the JAX (Bradbury et al., 2018) version from Lu et al. (2023), along with its PPO-GRU implementation. These tasks include card and board games where agents must recall previous observations to extract useful information for decision-making. For these tasks, the guider's observation is designed to include the critical information needed to remember, theoretically minimizing the imitation gap as long as the GRU can store the necessary information. Although in practice, GRU models struggle to retain all information, especially in complex tasks, this setup allows us to use a larger KL-threshold or clipping parameter, enabling the guider to explore further and provide more valuable supervision. For GPO-clip, due to the asymmetry with large $\delta$, we replace the clip$(\frac{\mu}{\pi}, 1 - \delta, 1 + \delta)$ with clip$(\frac{\mu}{\pi}, \frac{1}{r}, r)$. Further details on the experimental settings are provided in Appendix E.

Fig. 3 shows the results on 15 POPGym tasks, where we compare GPO-penalty and GPO-clip to PPO-V and PPO. The general conclusion mirrors the results from previous subsection, where GPO-clip typically outperforms GPO-penalty, followed by PPO-V and PPO. Key insights include:

**1**. The superior performance of GPO-penalty indicate that the ability of the guider to explore further without diverging too much from the learner proves valuable in these memory-based tasks.

**2**. While PPO-V outperforms PPO, its performance improvement is less pronounced in memory-based tasks than in the MuJoCo domain. This suggests that using additional information in the value function benefits noisy tasks but provides less of an advantage in tasks requiring memory.

**3**. In tasks like *BattleshipMedium* and *CountRecallHard*, neither GPO-penalty nor GPO-clip exhibit superior performance. We attribute this to the fixed hyperparameters across all tasks, which might not be optimal for these specific challenges. Further verification is presented in the next subsection.

Overall, our methods demonstrate strong performance across the majority of tasks, providing an effective solution for memory-based problems.

### 4.4 ABLATIONS AND DISCUSSIONS

In this section, we dive deeper into GPO's performance through ablations and further discussions.

**Why do GPO-clip and GPO-penalty outperform other baselines?** We attribute the success of GPO to two primary factors: (1) effective RL training of the learner, and (2) effective supervision from the guider.

The effectiveness of the RL can be demonstrated in Fig. 4(a), where we compare GPO-ablation with PPO-V on the *Humanoid* task. GPO-ablation, as described in Table 3, is GPO-penalty without the supervision term, with the learner's RL coefficient set to 1. This setup trains the learner similarly to PPO-V, but using the data

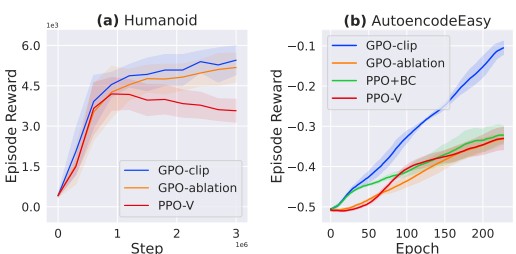

Figure 4: Ablation studies.

collected by the guider. From Fig. 4(a) where GPO-ablation outperforms PPO-V, we can conclude that the data collected by the guider better facilitates the learner's RL training. This demonstrates that GPO's ability to use a superior behavior policy improves the efficiency of the learner's RL training.

The effectiveness of the supervision comes from the guider being constrained to the "possibly good" region while still learning rapidly. This can be verified through experiments in Fig. 3 and Fig. 4(b). In these experiments, the RL term in GPO-clip was set to 0, meaning the learner was trained purely via supervision from the guider. In Fig. 4(b), we can observe that GPO-ablation, PPO+BC, and PPO-V perform similarly but lag behind GPO-clip. This shows that memory-based tasks benefit less from the learner's RL training. Since PPO+BC performs poorly in noisy tasks in Section 4.2 but comparably to PPO-V here, we can infer that supervision plays a particularly important role in tasks requiring memory. Additionally, the significant outperformance of GPO-clip over PPO+BC, even though both rely on pure supervision, suggests that GPO-clip's ability to constrain the guider's policy within the "possibly good" region is crucial to its success.

**Why does GPO-clip outperform GPO-penalty?** The primary the-
oretical difference between the two variants lies in how they regulate
the divergence between the guider and learner policies. GPO-clip
controls the total variation (TV) distance, while GPO-penalty con-
trols the KL divergence. GPO-clip only backtracks the guider when
it significantly diverges from the learner, whereas GPO-penalty con-
sistently pulls the policies back through a KL penalty, potentially
constraining the guider excessively. In Fig. 5, we can observe that
the KL divergence in GPO-penalty is consistently constrained near
$d_{\text{targ}} = 0.1$, whereas in GPO-clip, the KL divergence starts large
and decreases gradually. This suggests that GPO-clip allows the
guider more flexibility to explore further from the learner, providing
more effective supervision, while the distance constraint ensures
the learner eventually catches up. GPO-penalty, by consistently
regulating KL divergence, may over-constrain some policies and
under-constrain others, limiting its effectiveness.

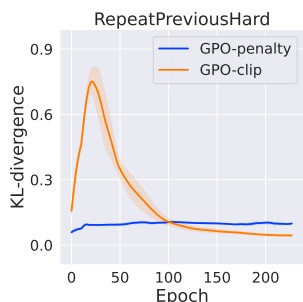

Figure 5: KL divergence dur-
ing training.

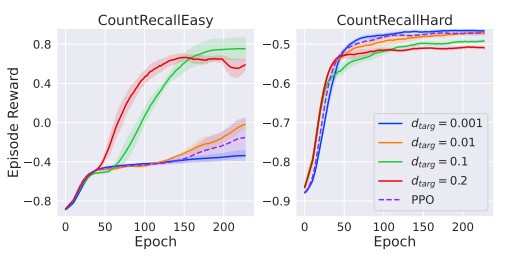

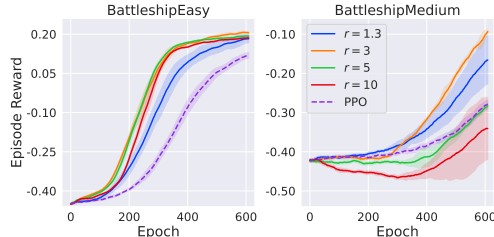

(a) GPO-penalty with different KL-threshold.      (b) GPO-clip with different clip parameter.

Figure 6: The results of GPO-penalty and GPO-clip with different hyperparameters.

**When does GPO fail**? One straightforward failure mode occurs when the guider learns slower
than directly training the learner via RL. This typically happens when the guider is provided with
inadequate information, which slows training instead of accelerating it. Another failure mode is
using inappropriate KL-threshold (clip parameters). For instance, in the *CountRecallHard* task
in POPGym, both GPO variants underperform compared to PPO and PPO-V. To explore this, we
conducted additional experiments (Fig. 6). In simpler tasks like *CountRecallEasy* and *BattleshipEasy*,
larger KL-thresholds (clip parameters) improve performance. However, for more difficult tasks
like *CountRecallHard* and *BattleshipMedium*, larger parameters lead to worse performance. This is
because harder tasks challenge memory models like the GRU used here. If the GRU cannot adequately
retain necessary information, the learner cannot follow the guider. In such cases, a large KL-threshold
(clip parameter) exceeds the "possibly good" region for learner, leading to an irrecoverable imitation
gap.

**How to set the KL-threshold/clip parameter?** As seen in the previous experiments, the configura-
tion of these hyperparameters depends on how large the "possibly good" region is. More specifically,
it depends on how well the learner's observation $o_l$ can infer the guider's observation $o_g$. In noisy
tasks, where $o_g$ cannot be easily inferred from noisy $o_l$, a smaller KL-threshold (clip parameter)
works best. In memory tasks, where $o_g$ can be predicted by $o_l$ given a memory model, a larger
KL-threshold (clip parameter) depending on the ability of the model is preferable.

## 5 CONCLUSION AND FUTURE WORK

In this paper, we introduced GPO, a method designed to leverage additional information in POMDPs
during training. Our experimental results demonstrate that the proposed algorithm effectively ad-
dresses noisy and memory-based partially observable tasks, offering a novel approach to utilizing
auxiliary information for more efficient learning. Future work could explore extending guided policy
optimization to the multi-agent setting, where agents often have access to global information during
training but are constrained to local observations during execution.

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

## A   RELATED WORKS

Leveraging additional information to accelerate learning in POMDPs has been explored across various frameworks and application domains (Vapnik & Vashist, 2009; Lambert et al., 2018; Lee et al., 2023). A prominent line of research focuses on Imitation Learning (IL), where expert knowledge, often equipped with extra information, significantly enhances performance in practical domains like autonomous driving (Bansal et al., 2018; De Haan et al., 2019) and robot navigation and planning (Choudhury et al., 2017; Bhardwaj et al., 2017). However, traditional IL methods such as Behavioral Cloning (BC) (Pomerleau, 1991; Torabi et al., 2018) and DAgger (Ross et al., 2011) often lead to sub-optimal solutions in scenarios requiring active information gathering by the agent (Pinto et al., 2018; Warrington et al., 2020). To overcome these limitations, recent research has focused on hybrid approaches that integrate RL with IL, often in the context of policy distillation (Czarnecki et al., 2019). For instance, Nguyen et al. (2022)modifies Soft Actor Critic (SAC) (Haarnoja et al., 2018) by replacing the entropy term with a divergence measure between agent and expert policies at each visited state. Similarly, Weihs et al. (2024) introduces a balancing mechanism between BC and RL training, adjusting based on the agent's ability to mimic the expert. Additionally, Walsman et al. (2023) applies potential-based reward shaping (Ng et al., 1999) using the expert's value function to guide the agent's policy gradient, while Shenfeld et al. (2023b) augments entropy in SAC to blend task reward with expert guidance, where the balance is based on the agent's performance relative to a reward-only learner. Despite these advances, expert-driven approaches often assume access to a reliable expert, which may not be feasible when only supplementary information is available. This has led to a growing body of work on co-training approaches where the expert and agent are learned jointly, with the expert conditioned on additional information. For example, Salter et al. (2021) proposes training separate policies for the agent and expert using spatial attention for image-based RL, aligning attention mechanisms through shared experiences. Song et al. (2020) co-trains two policies, each conditioned on different information, and selects the most successful rollouts from both policies to guide subsequent learning via RL or IL. Warrington et al. (2020) further develops this idea in adaptive asymmetric DAgger (A2D), where the expert is continuously refined through RL while supervising the agent. Beyond expert-based methods, a complementary approach involves embedding supplementary information directly into the value function within the actor-critic framework (Pinto et al., 2018; Andrychowicz et al., 2020; Baisero & Amato, 2021). This approach is particularly useful in multi-agent settings where global information is naturally accessible (Foerster et al., 2018; Lowe et al., 2017; Yu et al., 2022). In our experiments, we benchmark against several algorithms inspired by these lines of work, with detailed descriptions of the baselines provided in Appendix E.1.

## B   OMITTED PROOFS

**Proposition 1.** *If the guider's policy is updated using policy mirror descent in each GPO iteration:*

$$\hat{\mu} = arg\min\{-\eta_k\langle\nabla V(\mu^{(k)}), \mu\rangle + \frac{1}{1-\gamma}D_{\mu^{(k)}}(\mu, \mu^{(k)})\}, \tag{11}$$

*then the learner's policy update follows a constrained policy mirror descent:*

$$\pi^{(k+1)} = arg\min_{\pi\in\Pi}\{-\eta_k\langle\nabla V(\pi^{(k)}), \pi\rangle + \frac{1}{1-\gamma}D_{\pi^{(k)}}(\pi, \pi^{(k)})\} \tag{12}$$

*Proof.* First, since $D$ is a weighted sum of KL divergence, it satisfies the definition of a Bregman divergence. Therefore, for any distributions $p, q \in \Delta(A)^{|S|}$, we have

$$D_q(p, q) = h_q(p) - h_q(q) - \langle\nabla h_q(q), p - q\rangle, \tag{13}$$

where $h_q(p) = \sum_{s\sim d_q} p_s\log p_s$ is the negative entropy weighted by the state distribution.

Next, by backtracking $\mu^{(k)}$ to $\pi^{(k)}$ from the last time step, we get:

$$\hat{\mu} = \arg\min\left\{-\eta_k\langle\nabla V(\mu^{(k)}), \mu\rangle + \frac{1}{1-\gamma}D_{\mu^{(k)}}(\mu, \mu^{(k)})\right\}$$

$$= \arg\min\left\{-\eta_k\langle\nabla V(\pi^{(k)}), \mu\rangle + \frac{1}{1-\gamma}D_{\pi^{(k)}}(\mu, \pi^{(k)})\right\} \tag{14}$$

$$= \arg\min\left\{-(1-\gamma)\eta_k\langle\nabla V(\pi^{(k)}), \pi\rangle + h_{\pi^{(k)}}(\pi) - \langle\nabla h_{\pi^{(k)}}(\pi^{(k)}), \pi\rangle\right\},$$

The optimality condition for $\hat{\mu}$ requires:

$$-(1-\gamma)\eta_k \nabla V(\mu^{(k)}) + \nabla h_{\mu^{(k)}}(\hat{\mu}) - \nabla h_{\mu^{(k)}}(\mu^{(k)}) = 0, \tag{15}$$

where we use the fact that:

$$\nabla_p D_q(p,q) = \nabla_p h_q(p) - \nabla_p h_q(q). \tag{16}$$

Now, consider the update of the learner's policy, which involves a Bregman projection $\mathcal{P}_\Pi$:

$$
\begin{aligned}
\pi^{(k+1)} = \mathcal{P}_\Pi(\hat{\mu}) &= \arg\min_{\pi \in \Pi} D_{\mu^{(k)}}(\pi, \hat{\mu}) \\
&= \arg\min_{\pi \in \Pi} \big\{ h_{\mu^{(k)}}(\pi) - \langle \nabla h_{\mu^{(k)}}(\hat{\mu}), \pi \rangle \big\} \\
&= \arg\min_{\pi \in \Pi} \big\{ h_{\pi^{(k)}}(\pi) - \langle \nabla h_{\pi^{(k)}}(\pi^{(k)}) + (1-\gamma)\eta_k \nabla V(\pi^{(k)}), \pi \rangle \big\} \\
&= \arg\min_{\pi \in \Pi} \big\{ -(1-\gamma)\eta_k \langle \nabla V(\pi^{(k)}), \pi \rangle + h_{\pi^{(k)}}(\pi) - \langle \nabla h_{\pi^{(k)}}(\pi^{(k)}), \pi \rangle \big\} \\
&= \arg\min_{\pi \in \Pi} \big\{ -\eta_k \langle \nabla V(\pi^{(k)}), \pi \rangle + \frac{1}{1-\gamma} D_{\pi^{(k)}}(\pi, \pi^{(k)}) \big\}
\end{aligned} \tag{17}
$$

This completes the proof. $\qquad\square$

**Proposition 2.** *For policy $\pi$, $\mu$, $\beta$ and all state $s$, suppose $D_{TV}(\mu(\cdot|s), \beta(\cdot|s)) \lesssim \epsilon/2$, then we have*

$$\mathbb{E}_{a\sim\beta}\big[|1 - r^\pi(s,a)|\big] \lesssim \epsilon + \sqrt{2d_{targ}}. \tag{18}$$

*Proof.* First, let's examine the assumption $D_{TV}(\mu(\cdot|s), \beta(\cdot|s)) \lesssim \epsilon/2$ to check its validity.

Notice that at the start of each PPO policy update, the importance sampling ratio $r^\mu(s,a)$ equals 1 because the behavioral policy is equal to the policy being updated, i.e., $\beta(a|s) = \mu(a|s)$.

As PPO proceeds, $r^\mu(s,a)$ is updated multiple times using the same batch of samples. Due to the clipping function applied to $r^\mu(s,a)$, i.e., clip($r^\mu(s,a), 1-\epsilon, 1+\epsilon$), only state-action pairs for which $r^\mu(s,a) \in (1-\epsilon, 1+\epsilon)$ get updated. Hence, in the early epochs of PPO, with a properly tuned step size, we expect:

$$|1 - r^\mu(s,a)| \lesssim \epsilon. \tag{19}$$

Now, recalling the definition of total variation (TV) distance:

$$D_{TV}(\mu(\cdot|s), \beta(\cdot|s)) = \frac{1}{2}\sum_a |\mu(a|s) - \beta(a|s)| = \frac{1}{2}\sum_a \beta(a|s)|r^\mu(s,a) - 1| \lesssim \epsilon/2. \tag{20}$$

This confirms that the assumption $D_{TV}(\mu(\cdot|s), \beta(\cdot|s)) \lesssim \epsilon/2$ is reasonable, especially for the first few policy updates.

By the triangle inequality for total variation distance:

$$D_{TV}(\pi(\cdot|o), \beta(\cdot|s)) \leq D_{TV}(\pi(\cdot|o), \mu(\cdot|s)) + D_{TV}(\mu(\cdot|s), \beta(\cdot|s)), \tag{21}$$

we have

$$
\begin{aligned}
D_{TV}(\pi(\cdot|o), \beta(\cdot|s)) &\leq \sqrt{\frac{1}{2}D_{\mathrm{KL}}(\pi(\cdot|o), \mu(\cdot|s))} + D_{TV}(\mu(\cdot|s), \beta(\cdot|s)) \\
&\lesssim \sqrt{\frac{1}{2}d_{\mathrm{targ}}} + \epsilon/2,
\end{aligned}
$$

where we use Pinsker's inequality to bound the total variation distance between $\pi$ and $\mu$ in terms of their KL divergence.

Finally, since total variation is linked to the expected difference between probabilities under different policies, we have:

$$\mathbb{E}_{a\sim\beta}\big[|1 - r^\pi(s,a)|\big] = 2D_{TV}(\pi(\cdot|o), \beta(\cdot|s)) \lesssim \epsilon + \sqrt{2d_{targ}}. \tag{22}$$

This result implies that, under the assumption, the majority of samples are valid for updating the learner's policy during the early PPO epochs. $\qquad\square$

---

**Algorithm 1** Guided Policy Optimization

---

1: Input: initial policy parameters $\theta_0$, initial value function parameters $\phi_0$.
2: **for** $k = 0, 1, 2, ...$ **do**
3:     Collect a set of trajectories $\mathcal{D}_K = \{\tau_i\}$ by running guider's policy $\mu_k = \mu(\cdot|o_g; \theta_k)$ in the environment.
4:     Compute rewards-to-go $\hat{R}_t$.
5:     Compute advantage estimates $\hat{A}_t$ using GAE, based on the current value function $V_{\phi_k}$.
6:     Update policy parameter $\theta_k$ to $\theta_{k+1}$ by maximizing the GPO-penalty objective (9) or the GPO-clip objective (10).
7:     Fit value function by regression on mean-squared error:

$$\phi_{k+1} = \arg\min_\phi \frac{1}{|\mathcal{D}_K|T} \sum_{\tau \in \mathcal{D}_K} \sum_{t=0}^{T} \left( V_{\phi_k}((o_g)_t) - \hat{R}_t \right). \tag{23}$$

8: **end for**=0

---

## C    PSEUDO CODE

In this section, we present the pseudo code of our algorithm (see Algorithm 1). The algorithm is based on PPO, with an additional objective to leverage the extra information available during training.

## D    GPO ON TIGERDOOR-ALT PROBLEM

| state \ action | $a_L$ | $a_R$ |
|:---:|:---:|:---:|
| $s_L$ | 2 | 0 |
| $s_R$ | 0 | 1 |

Table 4: TigerDoor-alt problem

Here we provide an intuitive example to show how GPO can achieve optimal in the TigerDoor-alt problem. Initially, the guider's policy is uniform:

$$\mu(\cdot|s_L) = \mu(\cdot|s_R) = (0.5, 0.5)$$

After an update step, the guider's policy shifts to reflect the reward structure. For instance:

$$\mu(\cdot|s_L) = (0.7, 0.3), \;\; \mu(\cdot|s_R) = (0.4, 0.6)$$

The key here is that the higher reward for $(s_L, a_L)$ results in a larger gradient update compared to $(s_R, a_R)$ biasing $\mu(\cdot|s_L)$ more strongly toward $a_L$. Then the learner imitates the guider, resulting in:

$$\pi = \left( \frac{0.7 + 0.4}{2}, \frac{0.3 + 0.6}{2} \right) = (0.55, 0.45).$$

This adjustment brings the learner's policy closer to the optimal policy $(1, 0)$. Finally, after backtracking, the guider's policy is reset to match the learner:

$$\mu(\cdot|s_L) = \mu(\cdot|s_R) = (0.55, 0.45).$$

In subsequent iterations, this process continues with initial guider's policy $(0.55, 0.45)$, and result in the learner's policy gradually improving. For example, in the next iteration, we will observe: $\pi(a_L) > 0.55$ and $\pi(a_R) < 0.45$. This iterative refinement drives the learner toward the optimal policy.

The critical factor is that higher rewards for specific guider actions result in larger updates, which the learner captures through imitation. Simultaneously, the backtracking step ensures that the guider remains aligned with the learner, fostering consistent improvement.

# E  EXPERIMENTAL SETTINGS

## E.1  BASELINES

Here, we provide a brief introduction to the baselines used in the experimental section.

**PPO**. This is the standard algorithm used to train the learner without any extra information. The objective function is:

$$L(\pi) = \mathbb{E}\left[\min\left(r^\pi(o_l, a)A^\beta(o_l, a), r^\pi_{clip}(o_l, a, \epsilon)A^\beta(o_l, a)\right)\right], \tag{24}$$

where the behavioral policy is $\beta = \pi_{\text{old}}$.

**GPO-naive**. This is GPO-penalty without the auxiliary RL loss term. The objective is:

$$L_{\text{GPO-naive}}(\theta) = \mathbb{E}\left[\min\left(r^{\mu_\theta}A^\beta(o_g, a), r^{\mu_\theta}_{clip}A^\beta(o_g, a)\right) - \alpha D_{\text{KL}}\left(\mu_\theta(\cdot|o_l)||\pi_{\hat{\theta}}(\cdot|o_g)\right) \\ - D_{\text{KL}}\left(\mu_{\hat{\theta}}(\cdot|o_l)||\pi_\theta(\cdot|o_g)\right)\right]. \tag{25}$$

**GPO-ablation**. This is GPO-penalty without the BC loss term. The objective is:

$$L_{\text{GPO-ablation}}(\theta) = \mathbb{E}\left[\min\left(r^{\mu_\theta}A^\beta(o_g, a), r^{\mu_\theta}_{clip}A^\beta(o_g, a)\right) - \alpha D_{\text{KL}}\left(\mu_\theta(\cdot|o_l)||\pi_{\hat{\theta}}(\cdot|o_g)\right) \\ + \min\left(r^{\pi_\theta}A^\beta(o_g, a), r^{\pi_\theta}_{clip}A^\beta(o_g, a)\right). \tag{26}$$

**PPO-V**. This trains the learner using PPO, but with its value function taking $o_g$ as input. The objective is:

$$L(\pi) = \mathbb{E}\left[\min\left(r^\pi(o_l, a)A^\beta(o_g, a), r^\pi_{clip}(o_l, a, \epsilon)A^\beta(o_g, a)\right)\right]. \tag{27}$$

This method is a common approach to integrating additional information during training (Pinto et al., 2018; Andrychowicz et al., 2020; Baisero & Amato, 2021), especially in multi-agent settings (Foerster et al., 2018; Lowe et al., 2017; Yu et al., 2022). It can also be seen as an application of the potential-based reward shaping method (Walsman et al., 2023) with guidance from the value function of a training expert.

**ADVISOR-co**. This is a modified version of the ADVISOR algorithm (Weihs et al., 2024) since the original one does not involve the guider's training. The objective for guider is:

$$L(\mu) = \mathbb{E}\left[\min\left(r^\mu(o_g, a)A^\beta(o_g, a), r^\mu_{clip}(o_g, a, \epsilon)A^\beta(o_g, a)\right)\right]. \tag{28}$$

ADVISOR uses a balancing coefficient $w$ between BC and RL training, based on the distance between the guider's policy $\mu$ and an auxiliary imitation policy $\hat{\pi}$:

$$L(\pi) = \mathbb{E}\left[w\text{CE}(\mu(\cdot|o_g), \pi(\cdot|o_l)) + (1-w)\min\left(r^\pi(o_l, a)A^\beta(o_l, a), r^\pi_{clip}(o_l, a, \epsilon)A^\beta(o_l, a)\right)\right],$$

where $w = exp(-\alpha D_{\text{KL}}(\mu(\cdot|o_g), \hat{\pi}(\cdot|o_l)))$ and CE means cross-entropy. This can be seen as GPO-penalty without the backtrack term and with a different $\alpha$ update schedule. However, without backtracking, $w$ will quickly diminish because the auxiliary policy cannot follow the guider, effectively reducing this approach to pure PPO training for the learner.

**PPO+BC**. In this method, the guider is trained using PPO:

$$L(\mu) = \mathbb{E}\left[\min\left(r^\mu(o_g, a)A^\beta(o_g, a), r^\mu_{clip}(o_g, a, \epsilon)A^\beta(o_g, a)\right)\right], \tag{29}$$

while the learner is trained using BC with the guider:

$$L(\pi) = -\mathbb{E}\left[D_{\text{KL}}\left(\mu(\cdot|o_g), \pi(\cdot|o_l)\right)\right]. \tag{30}$$

**A2D**. Adaptive Asymmetric DAgger (A2D) (Warrington et al., 2020) is closely related to GPO, as it also involves co-training both the guider and the learner. A2D uses a mixture policy $\beta(a|o_g, o_l) =$

$\lambda\mu(a|o_g) + (1-\lambda)\pi(a|o_l)$ to collect trajectories and train the expert $\mu$ with a mixed value function $V(o_g, o_l) = \lambda V^\mu(o_g) + (1-\lambda)v^\pi(o_l)$. The objective is:

$$L(\mu) = \mathbb{E}\left[\min\left(r^\mu(o_g, o_l, a)A^\beta(o_g, o_l, a), r^\mu_{clip}(o_g, o_l, a, \epsilon)A^\beta(o_g, o_l, a)\right)\right], \qquad (31)$$

while the learner is updated through BC:

$$L(\pi) = -\mathbb{E}\left[\mathrm{D_{KL}}\left(\mu(\cdot|o_g), \pi(\cdot|o_l)\right)\right] \qquad (32)$$

In practice, A2D often sets $\lambda = 0$ or anneals it quickly for better performance. When $\lambda = 0$, A2D is equivalent to GPO-naive without the backtrack step, and it uses the learner's behavioral policy $\pi$ instead of the guider's policy $\mu$. Although A2D implicitly constrains the guider's policy through the PPO clip mechanism (which prevents the guider's policy from deviating too far from the learner's behavioral policy), this is insufficient to replace the explicit backtrack step. As discussed in Section 3.3, the gap between $\mu$ and $\pi$ can accumulate if the learner fails to follow the guider. As a result, most samples will be clipped as training progresses, leading A2D to fail to train a strong guider.

### E.2 HYPERPARAMETERS

The experiments in Sections 4.1 and 4.3 use the same codebase from Lu et al. (2023). The hyperparameters for these experiments are listed in Table 5.

For the experiments in Section 4.2, we use the codebase from Freeman et al. (2021). We perform a hyperparameter search for the original versions of the tasks and then fix the same hyperparameters for the partially observable and noisy variants. The hyperparameter search is detailed in Table 6, and the selected hyperparameters for the experiments are provided in Table 7. Other fixed hyperparameters are listed in Table 8.

| Parameter | Value (TigerDoor) | Value (POPGym) |
|---|---|---|
| Adam Learning Rate | 5e-5 | 5e-5 |
| Number of Environments | 64 | 64 |
| Unroll Length | 1024 | 1024 |
| Number of Timesteps | 2e6 | 15e6 |
| Number of Epochs | 30 | 30 |
| Number of Minibatches | 8 | 8 |
| Discount $\gamma$ | 0.99 | 0.99 |
| GAE $\lambda$ | 1.0 | 1.0 |
| Clipping Coefficient $\epsilon$ | 0.2 | 0.2 |
| Entropy Coefficient | 0.0 | 0.0 |
| Value Function Weight | 1.0 | 1.0 |
| Maximum Gradient Norm | 0.5 | 0.5 |
| Activation Function | LeakyReLU | LeakyReLU |
| Encoder Layer Sizes | 128 | [128,256] |
| Recurrent Layer Hidden Size | - | 256 |
| Action Decoder Layer Sizes | 128 | [128,128] |
| Value Decoder Layer Sizes | 128 | [128,128] |
| KL Threshold $d$ | 0.001 | 0.1 (0.001 for CartPole) |
| Clip $r$ | 1.1 | 10 (1.2 for CartPole) |
| RL Coefficient $\alpha$ | 1 | 0 (1 for CartPole) |

Table 5: Hyperparameters used in TigerDoor and POPGym.

### E.3 ENVIRONMENT DESCRIPTIONS

We provide a brief overview of the environments used and the guider's observation settings.

**MuJoCo tasks and CartPole in POPGym**: For these tasks, velocities and angular velocities are removed from the learner's observation. Gaussian noise with standard deviations of 0.1, 0.2, and 0.3 is added to the observations, corresponding to the difficulty levels *Easy*, *Medium*, and *Hard*, respectively. The guider, however, has access to the noiseless observations and the removed velocities.

| Parameter | Value |
|---|---|
| Reward Scaling $r_s$ | [0.1, 1] |
| Discount $\gamma$ | [0.97, 0.99, 0.997] |
| Unroll Length $l$ | [5, 10, 20] |
| Batchsize $b$ | [256, 512, 1024] |
| Number of Minibatches $n$ | [4, 8, 16, 32] |
| Number of Epochs $e$ | [2, 4, 8] |
| Entropy Coefficient $c$ | [0.01, 0.001] |
| KL Threshold $d$ | [0.01, 0.001] |
| Clip $\delta$ | [0.1, 0.3] |
| RL Coefficient $\alpha$ | [0, 2, 3] |

Table 6: Sweeping procedure in the MuJoCo domain.

| Task | $r_s$ | $\gamma$ | $l$ | $b$ | $n$ | $e$ | $c$ | $d$ | $\delta$ | $\alpha$ |
|---|---|---|---|---|---|---|---|---|---|---|
| Ant | 0.1 | 0.97 | 5 | 1024 | 32 | 4 | 0.01 | 0.001 | 0.3 | 2 |
| Halfcheetah | 1 | 0.99 | 5 | 512 | 4 | 4 | 0.001 | 0.001 | 0.1 | 2 |
| Humanoid | 0.1 | 0.99 | 5 | 512 | 32 | 4 | 0.01 | 0.001 | 0.1 | 2 |
| HumanoidStandup | 0.1 | 0.99 | 5 | 256 | 32 | 8 | 0.01 | 0.001 | 0.3 | 3 |
| InvertedDoublePendulum | 1 | 0.997 | 20 | 256 | 8 | 4 | 0.01 | 0.001 | 0.1 | 0 |
| Swimmer | 1 | 0.997 | 5 | 256 | 32 | 4 | 0.01 | 0.001 | 0.3 | 3 |
| Walker2d | 1 | 0.99 | 5 | 512 | 32 | 4 | 0.01 | 0.001 | 0.1 | 2 |

Table 7: Adopted hyperparameters in the MuJoCo domain. Notations correspond to Table 6.

**Autoencode**: During the WATCH phase, a deck of cards is shuffled and played in sequence to the agent with the watch indicator set. The watch indicator is unset at the last card in the sequence, where the agent must then output the sequence of cards in order. The guider directly observes the correct card to be output at each timestep.

**Battleship**: A partially observable version of Battleship game, where the agent has no access to the board and must derive its own internal representation. Observations contain either HIT or MISS and the position of the last salvo fired. The player receives a positive reward for striking a ship, zero reward for hitting water, and negative reward for firing on a specific tile more than once. The guider has access to a recorder that tracks all previous actions taken by the agent.

**Count Recall**: Each turn, the agent receives a next value and query value. The agent must answer the query with the number of occurrences of a specific value. In other words, the agent must store running counts of each unique observed value, and report a specific count back, based on the query value. The guider directly observes the running counts at each timestep.

**Repeat Previous**: At the first timestep, the agent receives one of four values and a remember indicator. Then it randomly receives one of the four values at each successive timestep without the remember indicator. The agent is rewarded for outputting the observation from some constant k timesteps ago, i.e. observation $o_{t-k}$ at time $t$. The guider has direct access to the value $o_{t-k}$ at time $t$.

### E.4 ADDITIONAL FIGURES

Fig. 8 shows the reward curves of the experiments presented in Section 4.2.

Fig. 12 illustrates the performance influenced by the parameter sharing and zero padding. We can observe that parameter sharing can sometimes impair performance, particularly when the observation dimension is large. For instance, in the *HumanoidStandup* task, the observation dimension is 400, which challenges the expressive capacity of the network. Thus, the decision to share the policy network represents a trade-off between memory efficiency and performance.

### E.5 COMPUTATIONAL COST

In this section, we present the computational cost of GPO (both GPO-penalty and GPO-clip share the same cost), PPO-V, and pure rollouts across several environments. The results, shown in Table

| Parameter | Value |
|---|---|
| Adam Learning Rate | 3e-4 |
| Number of Environments | 2048 |
| Episode Length | 1024 |
| Number of Timesteps | 3e7 |
| GAE $\lambda$ | 0.95 |
| Clipping Coefficient $\epsilon$ | 0.3 |
| Activation Function | SiLU |
| Value Layer Sizes | [128, 128] |
| Policy Layer Sizes | [128, 128] |

Table 8: Common hyperparameters used in MuJoCo domain.

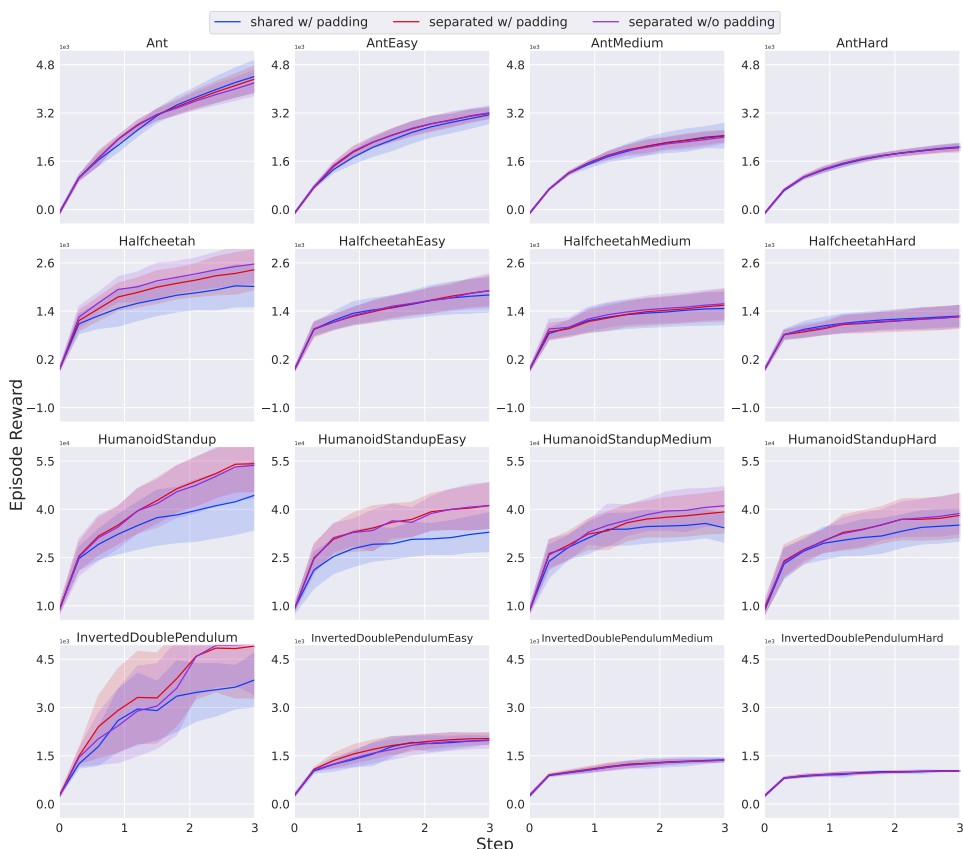

Figure 7: Comparing shared and separated policy networks of GPO-penalty.

9, , indicate that GPO is approximately 10% to 20% slower than PPO-V. It's important to note that GPO does not require any additional networks, highlighting its efficiency despite the slight increase in computational cost.

| Environment | GPO | PPO-V | Rollout Only |
|---|---|---|---|
| Ant | $1.19 \times 10^5$ | $1.36 \times 10^5$ | $4.23 \times 10^5$ |
| Halfcheetah | $6.27 \times 10^4$ | $7.21 \times 10^4$ | $2.55 \times 10^5$ |
| Humanoid | $6.29 \times 10^4$ | $7.18 \times 10^4$ | $2.50 \times 10^5$ |
| Swimmer | $3.33 \times 10^4$ | $3.83 \times 10^4$ | $1.50 \times 10^5$ |

Table 9: Frames per second (FPS) of GPO and PPO-V across several environments, computed on the NVIDIA GeForce RTX 4090.

## F  ADDITIONAL BASELINE RESULTS WITH A PRETRAINED TEACHER

This section presents results for SOTA teacher-student learning methods, including ADVISOR and TGRL, using a pretrained teacher in the Brax environment. The *Ant* task serves as an example, where the teacher is trained on the fully observable task using PPO. This pretrained teacher is then employed to train various algorithms on both the original fully observable task (*OriginalAnt*) and a partially observable version (*Ant*), consistent with the setup in Section 4.2.

The results are illustrated in Figure 9. In Figures 9(a) and (b), we can observe that in the *OriginalAnt* task (where the teacher was trained), teacher-student learning algorithms such as ADVISOR and TGRL significantly improve sample efficiency compared to baseline algorithms like PPO and SAC. However, Figures 9(c) and (d) reveal a contrasting outcome in the partially observable *Ant* task. Here, the teacher, being privileged, fails to provide meaningful supervision. As a result, ADVISOR and TGRL revert to their base algorithms, PPO and SAC. Additionally, PPO+BC does not degenerate into PPO due to the consistent BC loss, which adversely impacts its performance, making it worse than PPO.

Figure 10 further examines the KL divergence of these methods relative to the teacher they learned from. The results indicate that teacher-student algorithms effectively minimize KL divergence when the teacher is not privileged. However, when the teacher is inimitable, the mechanisms in ADVISOR and TGRL adjust (e.g., changing weights or coefficients) to prioritize their base RL algorithms, effectively discarding the teacher's influence.

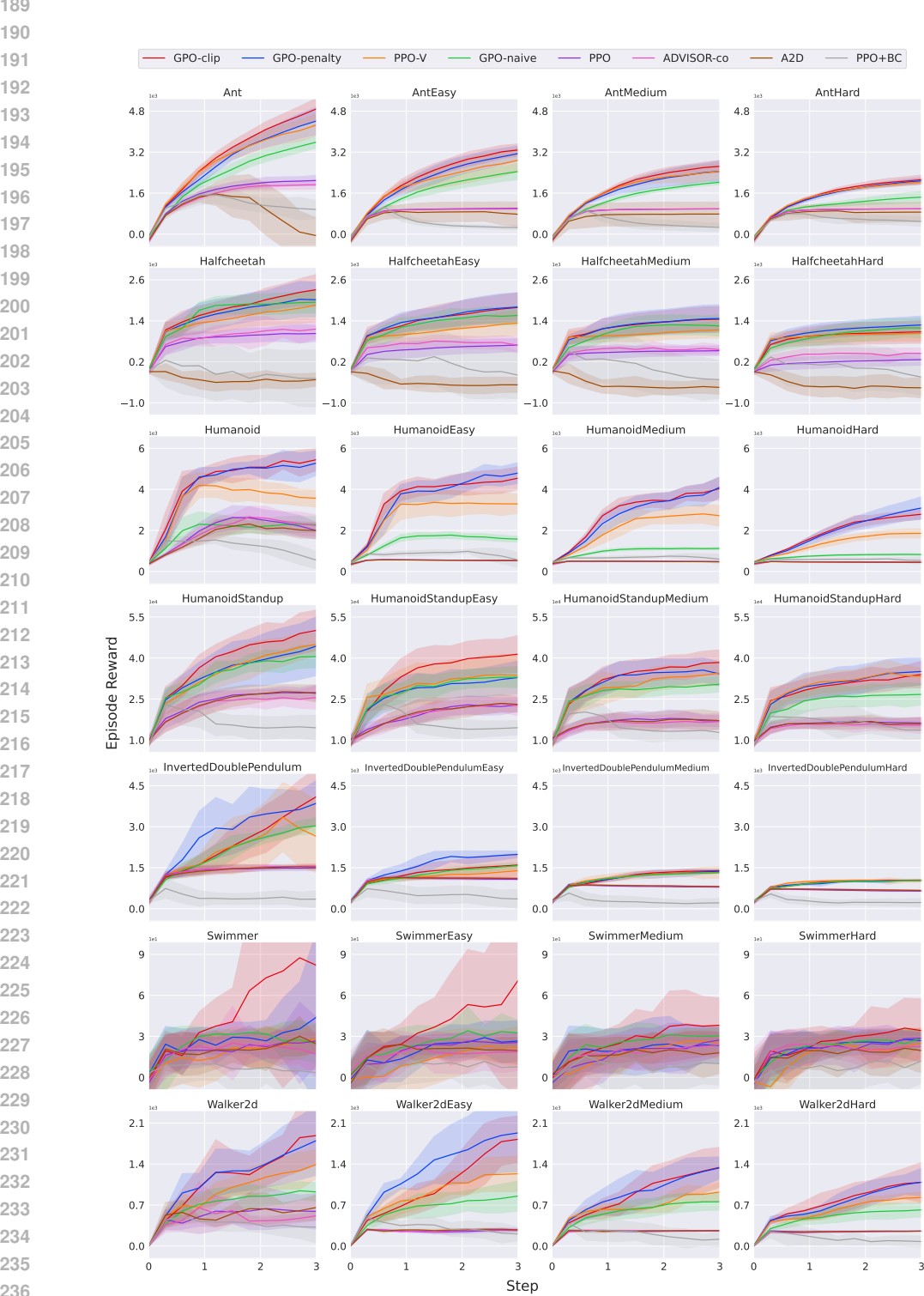

Figure 8: Comparing betweetn GPO and other baselines on 28 MuJoCo tasks.

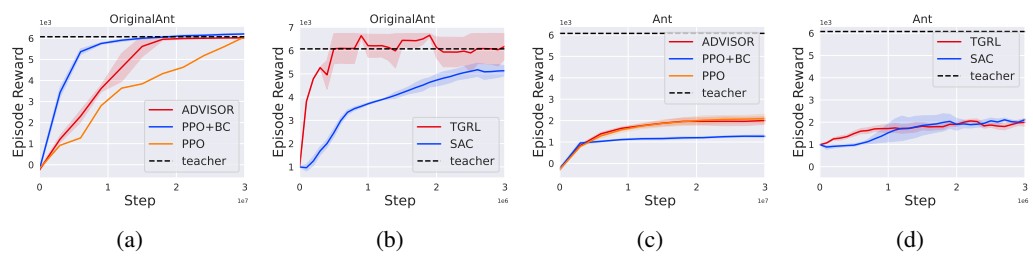

Figure 9: The results of PPO, PPO+BC, ADVISOR, SAC and TGRL.

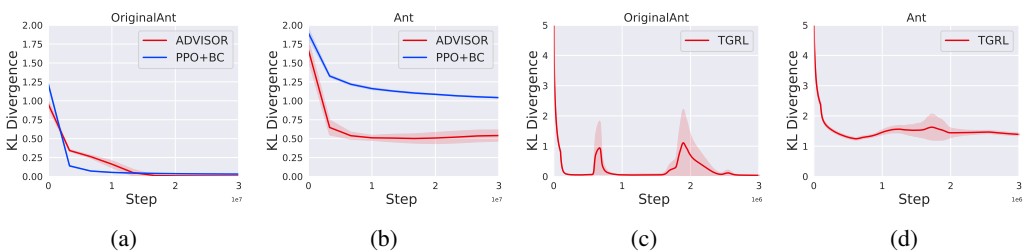

Figure 10: The KL divergence of PPO+BC, ADVISOR and TGRL to the teacher.

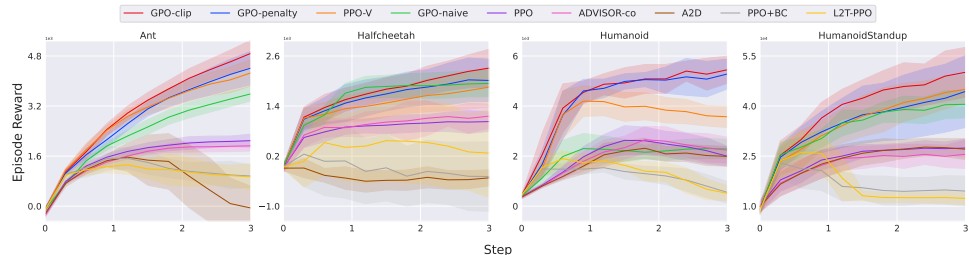

Figure 11: Comparing betweetn GPO and other baselines on 4 MuJoCo tasks

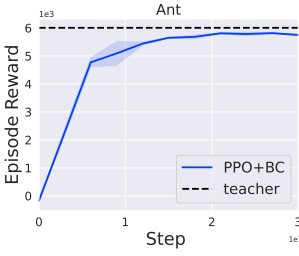

Figure 12: The results PPO+BC with a teacher pretrained by GPO.

.

