# OpenReview forum: "Leveraging Additional Information in POMDPs with Guided Policy Optimization"
_ICLR.cc/2025/Conference — Submitted to ICLR 2025_

### Official Review · Reviewer_Xvhn · 2024-10-29

**Soundness:** 1
**Presentation:** 3
**Contribution:** 2
**Rating:** 3
**Confidence:** 4

**Summary:**

In RL for POMDP, a common approach is to train a teacher with access to full observations using standard RL, and a student with access to partial observations to imitate the teacher. This paper proposes to apply guided policy search to teacher student learning, which limits the teacher training to be “similar” to the student during an incremental procedure that trains teacher and student together. This way, the co-training is “smoothed out”, and it is claimed to perform better on several POMDP benchmarks. Several algorithm variants based on PPO are proposed.

**Strengths:**

Strengths:
1. The guided policy search approach for POMDPs makes a lot of sense.
2. The paper was (mostly) clearly written.
3. Proposition 1 is a nice insight, and helps to understand the fundamental soundness of the approach.
4. The empirical comparisons between the different proposed GPO variants and their analysis is extensive.

**Weaknesses:**

Weaknesses:
1. The general idea of GPS for POMDP is not new, and has been proposed in [1], a reference and discussion that is missing.
2. The authors dismiss methods that first train a teacher policy and then a student policy using RL as “inefficient”, but I do not find this convincing enough without proper evaluation.
3. Following (2), the empirical comparison misses SOTA baselines that first train a teacher, like TGRL, and the authors modified the ADVISOR method to co-train the teacher and student together, which is not the intended use of ADVISOR and gives poor results.
4. The authors performed evaluations on several domains that are different from domains in prior work (e.g., ADVISOR, TGRL). As the method seems to be hyper parameter sensitive, a comparison with *previously published* results is necessary to understand its performance.
5. Some of the claims need to be made more formal.

Details on weaknesses:
1. The work in [1] already proposed the idea of GPS, where teacher policy has access to full state, and student is partially observable. Although the implementation there is very different (is was before PPO), the writing of this paper should be significantly modified to reflect the novelty based on that work. E.g., Line 50-53: This key insight has already been proposed in [1]. Also, I’m not familiar with more recent extensions of [1], but it has 100 citations, so an in-depth literature survey of this line of work is required.
2. In Line 44-45: The authors dismiss several recent works by claiming “typically assume access to a pre-trained teacher, which may not always be feasible. While one could train a teacher using additional information before training the agent, this two-step process is often inefficient and computationally expensive”. Following this, in their experiments, the authors do not include comparisons with methods that first train a teacher (TGRL, the original ADVISOR implementation) even though these are SOTA. I don’t understand why training a teacher first is claimed to be so inefficient. It indeed requires training a teacher policy, but that may be *easy* as the teacher is fully observable. There are additional factors that affect the learning time - if the method is more sensitive to hyper parameters, a denser sweep is required. If a method is more sample efficient, maybe running the teacher learning is not that costly. Moreover, even if training a teacher separately is costly, if it leads to better performance of the student then it still may be preferred. Fortunately, this can easily be evaluated by comparing with methods that separate teacher and student learning, and reporting the total number of learning iterations (or wall clock time, etc.). Ideally, this would be reported also on domains where previous methods were tested and optimized for, or on new domains, but including the cost of the hyper parameter sweep too.
3. The authors should add comparisons to the original ADVISOR, and also with PPO+BC where the teacher policy is first trained to convergence. In [2], a paper that is cited by the authors, the TGRL method significantly outperforms the ADVISOR baseline. Therefore, a comparison to TGRL should be added.
4. Importantly, as the algorithm seems to be sensitive to hyper parameters, an evaluation on domains where previous baselines were tested on, and their published results, is necessary to understand how much of the performance boost is due to hyper parameter tuning, and how much is due to the method (the authors evaluate on POPGym and noisy Mujoco, which is great, but different from previous work so impossible to directly answer this question). To emphasize this point: in several recent teacher-student algorithms for POMDPs (TGRL, ADVISOR, etc.) the key difficulty is balancing between several cost terms (teacher following, RL). From sections 3.2,3.3 it appears that a similar case holds here (though with different costs), and selecting tricks and hyper parameters that balance the costs well is critical. Since previous methods, such as TGRL, devised methods for automatically tuning the costs balance, an in-depth comparison with their results is relevant.
5. In Line 167-174: it would be better if this “hand wavy” paragraph is translated into a formal result. E.g., can you write “In other words, the update of the learner’s policy can inherit the properties such as monotonic policy improvement (Schulman et al., 2015a) from trust-region algorithms.” formally? Or formalize the claim “This suggests that GPO can effectively address challenges in IL, such as dealing with a suboptimal teacher or the imitation gap, while still framing the learner’s policy as being supervised by the guider”? In particular, previous works like TRPO considered fully observable policies, but here \pi is partially observable - what results exactly carry over to this case? Line 176-185: the explanation here is also vague. Why do “policy gradients for the learner suffer from high variance”? It would be better to limit the explanation to concrete statements. Also, if I understand correctly, the intuition explained here is also the intuition behind many previous works on teacher-student learning, and not specific to the current method.

Summary (and explanation for my score): There is clearly an interesting idea here, and optimizing the teacher policy to “align” with the student seems like it could help learning. That said, as the novelty here is only in the implementation (and not the general idea), this paper should be evaluated based on empirical comparisons to SOTA teacher-student methods for POMDPs, which in their present form, are not extensive enough. Taking in the required changes in the writing and the experiments, my impression is that a major	revision is needed for this paper to be publishable.

Other comments (do not affect score):

Line 145: in Guider Training step 2: based on the definition of V in line 76, the solution to max_mu V_mu is the optimal MDP policy, regardless of steps 1,3,4. So I don’t understand why it makes sense to iteratively compute this step. If the point is to make an incremental update, that starts from the policy in step 4, this should be clearly stated.

Line 208: L_1 is defined for mu, not pi. Is this a typo? Also, should it be L_3(mu) and not L_2(pi)?

References:

[1] Zhang, Marvin, et al. "Learning deep neural network policies with continuous memory states." 2016 IEEE international conference on robotics and automation (ICRA). IEEE, 2016.

[2] Shenfeld, Idan, et al. "Tgrl: An algorithm for teacher guided reinforcement learning." International Conference on Machine Learning. PMLR, 2023.

**Questions:**

see above.

---

> ### Author Response · Authors · 2024-11-18
>
> We appreciate the reviewer’s valuable feedback.
>
> 1. **The general idea of GPS for POMDP is not new, and has been proposed in [1]**.
>
> We acknowledge that the general idea of applying GPS to POMDP is not completely new and has been explored in prior work, including [1]. However, the **key insight** of our paper lies in keeping the teacher "possibly good" (aligned with the student). This insight is not specific to [1] but rather an inherent idea in all GPS methods. We adapt and **extend this principle to develop a novel RL training algorithm**.
>
> While [1] extends GPS by incorporating memory models, its approach inherits the same foundational ideas from the broader GPS methodology. Additionally, the application of GPS in partially observed domains has been suggested in earlier works. For instance, the original GPS paper [2] mentions that its controller is suitable for partially observed domains, and another GPS study [3] highlights its appeal for tasks with partial observability or information limitations.
>
> We do not claim that the idea is entirely original to our work. As noted in lines 49–50 and 136–137 of our paper, our method (GPO) is inspired by GPS, as reflected even in its naming. However, we recognize the need for a more comprehensive introduction to GPS-related work. This has been addressed in the revised Appendix A and in the general rebuttal above.
>
> 2&3. **Missing comparisons to SOTA methods where a teacher is trained first using full observation**.
>
> We appreciate this feedback and agree that these comparisons are important. We have included additional evaluations in response, as detailed in the general rebuttal above and Appendix E.
>
> 4. **Hyperparameter sensitivity and the need for comparisons with previously published results**.
>
> Our method is **not sensitive** to hyperparameters. Most hyperparameter tuning in our experiments is conducted for **PPO**, as it is sensitive in Brax, as suggested in https://github.com/google/brax/tree/main/datasets. Specifically:
> - We performed hyperparameter sweeps on **PPO** to optimize its performance for each task and applied these parameters consistently to **all baseline methods** that require PPO.
> - Our method, GPO-penalty, does not require any additional hyperparameter tuning. GPO-clip requires only two hyperparameters to be adjusted.
> - In POPGym, neither GPO-penalty nor GPO-clip necessitates any hyperparameter tuning.
>
> Regarding comparisons with methods like TGRL in their setting, we believe these are unnecessary since the result is straightforward. If an effective, pre-trained teacher is provided, methods like TGRL perform better as our method does not benefit from pre-trained teachers. However, in scenarios where only additional information is available, training a teacher first and then apply teacher-student methods will not be a good choice since the **trained teacher usually fails to provide useful supervision**. As a result methods like TGRL and ADVISOR typically degenerate into SAC or PPO. In contrast, our method effectively handles this situation and significantly outperforms base RL algorithms.
>
> 6. **Clarifying and formalizing claims**.
>
> **Line 167-174**: Partial observability does not impede RL convergence, as POMDPs can be reformulated into MDPs using belief states. While this transformation may affect optimality, it does not prevent convergence. This paragraph highlights that our method is theoretically equivalent to directly applying RL methods, thereby inheriting the properties of these methods. For instance, there is no need to reformulate Equation (3) into a TRPO-style update, as it already possesses the monotonic improvement property (see Lemma 7 of [4]). Consequently, when comparing our approach with IL methods, the comparison essentially becomes RL vs. IL. Notably, RL methods are not limited by a suboptimal teacher or the imitation gap.
>
> **Line 176-185**: Yes, it is the same intuition for teacher-student learning.
>
> **Line 145**: Yes, it is an incremental update. This has been revised in the updated paper.
>
> **Line 208**: Thanks. This was a typo, which has been corrected in the updated paper.
>
> [1] Zhang, Marvin, et al. "Learning deep neural network policies with continuous memory states." 2016 IEEE international conference on robotics and automation (ICRA). IEEE, 2016.
>
> [2] Levine S, Koltun V. Guided policy search. International Conference on Machine Learning. 2013.
>
> [3] Montgomery W, Levine S. Guided policy search as approximate mirror descent. arXiv preprint arXiv:1607.04614, 2016.
>
> [4] Lin Xiao. On the convergence rates of policy gradient methods. Journal of Machine Learning Research, 23(282):1–36, 2022.

---

> > ### Comment · Reviewer_Xvhn · 2024-11-18
> > **effective or not effective teacher**
> >
> > Thank you for your response and for the additional experiments.
> >
> > I don't understand the following claim:
> >
> > "However, in scenarios where only additional information is available, training a teacher first and then apply teacher-student methods will not be a good choice since the trained teacher usually fails to provide useful supervision. "
> >
> > The setting in previous work, e.g., TGRL, was a fully observable teacher and partially observable student. The teacher was trained using standard RL on the fully observed state. Still, it was shown to help the student learn, as in many domains, some of the teachers actions are still relevant and can be imitated, while some are not, and must be re-learned using RL. So the teacher still carries some useful information.
> >
> > The authors response "If an effective, pre-trained teacher is provided, methods like TGRL perform better" seems to imply that the teacher in TGRL was trained to solve the partially observed domain. But to the best of my knowledge that was not the case in TGRL/Advisor.
> >
> > So I still think that it's mandatory to compare with the partially observed domains in previous work.

---

> > > ### Author Response · Authors · 2024-11-18
> > >
> > > Thank you for your prompt reply.
> > >
> > > We understand the setting of TGRL, and our additional experiment involves exactly training the teacher using standard RL on the fully observed state. However, previous methods lack theoretical guarantees that such a trained teacher will be helpful. As we demonstrate, even in popular MuJoCo domains, there are cases where the teacher is ineffective, which suggests that pre-training a teacher may not always be a reliable approach. In contrast, our method provides a theoretical guarantee of the teacher's effectiveness and is more efficient as it does not require pre-training a teacher. We believe this underscores the significance and practical advantages of our approach.
> > >
> > > Additionally, there are important missing details that make reproducing TGRL's results challenging:
> > >
> > > 1. Environment Details:
> > > - While the Lava Crossing and Memory tasks seem to belong to the MiniGrid library, specific settings (e.g., grid size) are not provided.
> > >
> > > - We didn't find the Light-Dark Ant and Shadow Hand environments. There is no information about how noise is added to Light-Dark Ant or the exact observation setup for Shadow Hand.
> > >
> > > 2. Code Availability:
> > > TGRL reportedly uses DQN for Memory and Lava Crossing tasks, and an LSTM memory model is used, but these implementations are missing. Only an SAC implementation with an MLP version of TGRL is provided, and this simplifies the teacher coefficient update to a fixed value.
> > >
> > > The environments used in TGRL appear to be highly customized, whereas we believe the environments in our work are more widely recognized, reproducible, and available.

---

> > > > ### Comment · Reviewer_Xvhn · 2024-11-21
> > > > **followup questions**
> > > >
> > > > I have a couple of followup questions.
> > > >
> > > > - our method provides a theoretical guarantee of the teacher's effectiveness and is more efficient as it does not require pre-training a teacher.
> > > >
> > > > Can the authors explain what formal result shows that the teacher is effective (other than not hurting performance, which was also claimed in previous works)? Or that GPO  is more efficient than standard RL (or, e.g., previous works like TGRL)? I'm not sure I see this from Proposition 1.
> > > >
> > > > - an MLP version of TGRL is provided, and this simplifies the teacher coefficient update to a fixed value.
> > > >
> > > > What do you mean by "simplifies the teacher coefficient update to a fixed value"?

---

> > > > > ### Author Response · Authors · 2024-11-22
> > > > >
> > > > > 1. **The Effectiveness of the Teacher.**
> > > > >
> > > > > Since the term "effective teacher" was not explicitly defined in prior works, we propose the following definition based on properties that we believe an effective teacher should possess:
> > > > >
> > > > > (1) **Good performance**: The teacher's own policy demonstrates improvement, akin to the policy improvement property in standard RL, where: $V(\mu^{(k+1)})\ge V(\mu^{(k)})$.
> > > > >
> > > > > (2) **Imitability by the student**: The teacher's policy can be closely approximated by the student's policy, formally expressed as $\min_{\pi\in\Pi}D_{KL}(\mu||\pi)$ could be 0 or very small, given teacher’s policy $\mu$.
> > > > >
> > > > > A teacher trained purely with RL typically satisfies property (1) but may fail to meet property (2), as its policy might not be directly imitable by the student. In contrast, our method inherently satisfies property (2) because, at every iteration, it traces back the teacher's policy to ensure $\mu^{(k)}$ remains imitable for any $k$. To demonstrate the effectiveness of our teacher, we must show that despite the traceback operation, the teacher still achieves good performance.
> > > > >
> > > > > **Proof**:
> > > > >
> > > > > For brevity, we omit coefficients related to $\eta$ and $\gamma$.
> > > > >
> > > > > Let us consider the update of the teacher's policy at the kkk-th iteration in GPO:
> > > > >
> > > > > $$
> > > > > \hat{\mu}=arg\min_\mu[-<\nabla V(\mu^{(k)}),\mu>+D_{KL}(\mu||\mu^{(k)})],
> > > > > $$
> > > > >
> > > > > $$
> > > > > \pi^{(k+1)}=arg\min_{\pi\in\Pi} D_{KL}(\pi, \hat{\mu}), \mu^{(k+1)}=\pi^{(k+1)}.
> > > > > $$
> > > > >
> > > > > Thus, the update of $\mu$ can be written as:
> > > > >
> > > > > $ \mu^{(k+1)}=arg\min_{\mu\in\Pi}D_{KL}(\mu, \hat{\mu})$.
> > > > >
> > > > > This equation confirms property (2), as $\mu\in\Pi$ in GPO is always achievable by $\pi$.
> > > > > Next, consider the policy gradient theorem: $\nabla V(\mu)=d(\mu)Q(\mu)$. For the teacher, this implies:
> > > > >
> > > > > $\hat{\mu}=arg\min_\mu[-<Q(\mu^{(k)}),\mu>+D_{KL}(\mu||\mu^{(k)})],$
> > > > >
> > > > > which has the closed-form solution: $\hat{\mu}=\mu^{(k)}exp(Q(\mu^{(k)}))/z^{(k)}$, where $z$ is the partition function.
> > > > > From the definition of $\mu^{(k+1)}$, we have
> > > > >
> > > > > $D_{KL}(\mu^{(k+1)}, \hat{\mu})\le D_{KL}(\mu^{(k)}, \hat{\mu})$
> > > > >
> > > > > implying:
> > > > >
> > > > > $E_{\mu^({k})}[\log z^{(k)}-Q(\mu^{(k)})]\ge E_{\mu^({k+1})}[\log z^{(k)}-Q(\mu^{(k)})+\log(\mu^{(k+1)}/\mu^{(k)})]$.
> > > > >
> > > > > Therefore,
> > > > >
> > > > > $E_{\mu^{(k+1)}}[Q(\mu^{(k)})] - E_{\mu^{(k)}}[Q(\mu^{(k)})]\ge D_{KL}(\mu^{(k+1)}||\mu^{(k)})\ge 0.$
> > > > >
> > > > > Using the performance difference lemma, we conclude: $V(\mu^{(k+1)})- V(\mu^{(k)})\ge 0$, which completes the proof.
> > > > >
> > > > > Thus, our GPO method ensures the teacher's imitability while maintaining the policy improvement property of standard RL. Although we cannot theoretically derive that GPO is always more efficient than standard RL (as its efficiency depends on the quality of privileged knowledge), we think this has been validated empirically.
> > > > >
> > > > > In addition, in Figure 12, we add an experiment that compares the teacher trained with our method to the teacher trained by pure RL under **identical global observations**. Comparing figure 12 to figure 9(c), our teacher provides effective guidance, substantiating our claims.
> > > > >
> > > > > 2. **Details about TGRL.**
> > > > >
> > > > > In TGRL, the teacher coefficient update is expected to be proportional to the performance difference, as described in the original paper. However, the code implements a fixed update value, stating that this modification stabilizes training. Based on our understanding, TGRL may be sensitive to how $J(\pi)$ is measured.
> > > > >
> > > > > The implementation uses a _history-length_ of rewards to estimate $J(\pi)$ via averaging.
> > > > > In unstable environments, a short history length causes $J(\pi)$ to vary significantly, leading to unstable performance. Conversely, a long history length may result in suboptimal updates, as the teacher coefficient adjusts too slowly and is influenced by outdated performance metrics.

---

> > > > > > ### Comment · Reviewer_Xvhn · 2024-11-24
> > > > > > **followup**
> > > > > >
> > > > > > Thank you for answering all my questions.
> > > > > >
> > > > > > I agree that your results can show that the teacher improves (and I noted the nice Prop. 1 in my original review). However, what we ultimately care about is the student (as that's what you evaluate in the experiments), and there is no theoretical result that shows that your student is learning faster than alternative methods or pure RL (I assume that obtaining such results is really not trivial, as POMDPs are NP Hard, so in the worst case, the student would have to learn as slowly as pure RL).
> > > > > >
> > > > > > Indeed, your empirical results show that GPO outperforms pure RL on some domains, but so did previous work (TGRL, Advisor, etc.), so I don't see the theoretical statements above as anything that obviates an extensive comparison.
> > > > > >
> > > > > > Ultimately, GPO falls in exactly the same problem setting as teacher student methods like TGRL (actually, GPO is even more constrained, as the teacher has to be trained in a specific way, while in TGRL the teacher training is arbitrary). Therefore, *GPO must be evaluated in comparison to previously published SOTA results for this work to be publishable*. We may learn that GPO works better on some (or all domains), but importantly, we should understand whether the benefits of GPO and TGRL/Advisor are complementary (expoiting different advantages of the teacher), which may drive further research.
> > > > > >
> > > > > > I understand that reproducing TGRL is not as immediate as a python import statement, but reproducing previous work is part of standard scientific practice, and typically involves writing code, some experimentation, contacting the corresponding authors, etc.

---

> ### Author Response · Authors · 2024-11-25
>
> Thank you very much for your time and detailed review. We’d like to take this opportunity to clarify a few points further:
>
>
> > There is no theoretical result that shows that your student is learning faster.
>
> You are correct that the theoretical results for GPO only ensure that the student has the same guarantees as pure RL, rather than proving greater efficiency. This is because, in the worst-case scenario, the additional information provided by the teacher may be ineffective, and GPO’s learning rate would align with that of pure RL. It is not feasible to establish a universal theoretical result proving faster learning in all cases for this reason.
>
> However, it is worth noting that no theoretical result demonstrates that TGRL or ADVISOR performs at least as well as pure RL either.
>
> > GPO must be evaluated in comparison to previously published SOTA results for this work to be publishable.
>
> We appreciate this point and would like to highlight that GPO has already been compared against TGRL and ADVISOR on MuJoCo tasks. As shown in Figures 9(c) and 9(d), TGRL and ADVISOR achieve approximately 2e3, whereas GPO-clip and GPO-penalty (Figure 2 or Figure 8) achieve around 4.5e3—without even accounting for the teacher’s training cost.
>
> It is evident that GPO is less competitive when an effective teacher is available. However, GPO demonstrates significant advantages when directly training a teacher offers no benefit. The important question is: how frequently does this latter scenario occur? Our experiments suggest that this is indeed common in MuJoCo tasks, which is one of the most popular domains, demonstrating both the universality of the problem and the applicability of our method.
>
> We do not understand why it is necessary to compare GPO in less popular domains proposed by previous work. Even if GPO cannot outperform TGRL or ADVISOR in all their experiments, this does not undermine our primary contribution, which focuses specifically on scenarios where directly training a teacher is ineffective. Moreover, it is worth noting that even TGRL and ADVISOR do not rely on identical domains across their evaluations.

---

> > ### Comment · Reviewer_Xvhn · 2024-11-25
> > **response**
> >
> > It is necessary because the claim that your method beats the SOTA is potentially a very important contribution.
> >
> > Us reviewers need to evaluate this claim. How can we tell if your method beats the SOTA because it is really a better algorithm, or because of some suboptimality in your baseline implementation?
> >
> > Evlauating against published results would give a clear answer.

---

> > > ### Author Response · Authors · 2024-11-26
> > >
> > > As we discussed earlier, reviewers can evaluate our paper under the assumption that our method underperforms SOTA methods in all their environments. We believe this does not undermine our main contribution, as our focus is on scenarios where the teacher is less effective, and we demonstrate that our method outperforms the alternatives in more popular domains.

---

> > > > ### Author Response · Authors · 2024-11-28
> > > >
> > > > While we agree with many of the reviewer’s points, we respectfully disagree that a paper must be evaluated on previously proposed, non-open-sourced environments to be publishable. Consider the following:
> > > > - If a prior paper does not release its environment and custom settings, should all subsequent papers be rejected for not using them?
> > > > - TGRL, for instance, does not evaluate on the same domains as ADVISOR. Furthermore, it does not even use the same version of ADVISOR (which is on-policy). This raises a similar question: “How can we tell if TGRL beats the SOTA because it is really a better algorithm, or because of some suboptimality in their baseline implementation?”

---

### Official Review · Reviewer_ReJ3 · 2024-11-02

**Soundness:** 3
**Presentation:** 3
**Contribution:** 2
**Rating:** 6
**Confidence:** 3

**Summary:**

This paper introduces a method to co-train a guider and a learner. The former has access to the full state of the environment, while the learner only has access to a partial observation of the environment state. The guider commences learning in the environment from scratch, and is updated to maximise the objective while ensuring closeness with the learner policy. The learner policy is trained using a combined behaviour cloning and RL loss, based on the data collected by the guide.

**Strengths:**

1. Effectively simplifies learning in a noisy or partially observable environment, compared to learning from scratch
2. Thorough evaluation of the algorithm on many different kinds of environments, and good comparisons to other algorithms

**Weaknesses:**

1. I am not convinced as to the usefulness of this algorithm, if you truly have a task where a simulation represents a perfect noiseless and fully-observed version of the task, then why not just train a policy exclusively on the simulation, then for real-life application, just learn a map (i.e. your $f(s)$ from the partial to the full observations using supervised learning? This algorithm requires simultaneous access to the full and partial observation during training, so creating that map should be reasonably straightforward. You could then pass the full observation onto the policy trained on the simulation. Would this not be fully optimal? I’m not sure if I’m missing something here.
2. In Section 2.2, line 130 states “without requiring RL training for the learner”, by introducing the RL objective, it seems to be essentially RL, so it would have been good to see a comparison with a straight learner on the unaugmented noisy observations, just to verify the zero-padding was not slowing learning.
3. Would be good to reference the proof of Proposition 1 contained in the Appendix in the main text.

Minor
Line 201: missing “.”
Line 374: “betweetn”

**Questions:**

1. How would you foresee that this approach is different to, or superior to, my suggestion in Weakness 1.?
2. Shouldn't the the argmin in 2 and 3 should be an argmax, and only around the first term? In (Xiao, 2022), I believe they use an argmin in Equation 5 because they have swapped reward for regret in the value function, but it is not clear to me if you have done that also - it seems in item 2 in the list under Section 3.1, you state that you wish to maximise the objective.
3. Is the proof for Proposition 1 still relevant, given the final algorithm also includes the RL loss?

---

> ### Author Response · Authors · 2024-11-18
>
> We appreciate the reviewer’s feedback.
>
>
> 1. **Why not learn a map from partial to the full observations**？
>
> In general, a map that can recover true information from partial or noisy data does **not exist** in most cases. For example, it is not possible to derive global coordinates solely from local coordinates or to infer instantaneous speed from position histories (only average speed can be recovered). Similarly, if we aim to train a function $f$ to map noisy observations to true observations, i.e., $f(s+normal(0,\sigma))\to s$, regression typically yields an identity map $f$.
>
> Even if recovering true information were theoretically feasible, it would compromise the robustness of the resulting policy. Recovered observations would inevitably differ from the original, leading to performance degradation.
> For instance, in the _Ant_ task, a policy achieving 6000 episode rewards under noise-free conditions drops to 5000 with normal(0,0.02) noise and to 1000 with normal(0,0.1) noise. In contrast, our method maintains a reward of 3000 under normal(0,0.1) noise.
>
>
> 2. **Explanation for line 130: “without requiring RL training for the learner” and zero-padding impact on learning speed**.
>
> Our statement in line 130 aligns with the results in Section 4.1, where **GPO-naive achieves optimal performance without requiring RL training for the learner**. This supports the optimality guarantee introduced in Proposition 1.
>
> Additionally, zero-padding does not slow learning, as demonstrated in the experiments included in the figure 7 in Appendix.
>
> 3. **Reference to Proposition 1 in the main text**.
>
> Thanks. We have added a reference in the main text.
>
> 4. **Should the argmin in (2) and (3) be argmax, and only around the first term**?
>
> Thank you for identifying this issue. There was indeed an inconsistency in the notation. We have corrected it in the revised paper.
>
> 5. **Relevance of Proposition 1 given the inclusion of RL loss in the final algorithm**.
>
> Yes. Proposition 1 tells that GPO update is equivalent to direct RL update to the learner. The final algorithm apply GPO+RL update, which is also equivalent to RL update to the learner. Specifically, Proposition 1 tells that $\pi^*=arg\min J_{GPO}(\pi)= arg\min J_{RL}(\pi)$. The final algorithm $\pi^*=arg\min (J_{GPO}(\pi)+ J_{RL}(\pi))= arg\min J_{RL}$, since $\min(J_{GPO}(\pi)+ J_{RL}(\pi))\ge \min J_{GPO}(\pi)+ \min J_{ RL }(\pi)$ and the equality holds only when $\pi=\pi^*$.

---

> > ### Comment · Reviewer_ReJ3 · 2024-11-24
> >
> > Thank you for the clarification regarding points 2. and 5.
> >
> > As for 1., I am still confused. Isn't the backtracking term $L_3(\mu)=E[D_{KL}(\mu(\cdot|s),\pi(\cdot|o)]$ essentially mapping $o$ to $s$? For example, you say it is impossible to derive global coordinates from local coordinates, but given that your training method requires simultaneous access to global and local observations, why could you not use that to learn what global coordinates the local coordinates should map to, using an MSE loss similar to $L_3$?
> >
> > I am also not sure if your Ant example describes what I am suggesting. You say the Ant example trained under noise-free setting achieving 6000 reward drops to 1000 under normal(0,0.1) noise. This makes sense, as it has no idea how to act under these noisy observations. To clarify, I am instead suggesting that while training the policy $\mu$ using noise-free observations, e.g. $s_t$, you also learn a map $f(o,s)$ from the partial observation $o_t$ to the noise-free state $s_t$ using MSE loss. Then, following learning, a noisy $o_t$ observation will first pass through $f(o,s)$, producing $s_t$, which is then passed onto the trained $\mu$. This seems more straightforward than the approach you propose.
> >
> > I am happy to accept that I have a fundamental misunderstanding here, however I need further clarification as described above from your initial response. Thank you.

---

> > > ### Author Response · Authors · 2024-11-24
> > >
> > > Thank you for your reply.
> > >
> > > We’d like to clarify our point further. The central idea is that global observations typically contain more information than can be deduced from local observations. Here are two key examples:
> > > - **Global Coordinates**:
> > > Consider a scenario where the global coordinates are $(x_1, x_2)$, and the corresponding local coordinate is $y=x_1-x_2$. Ideally, one might seek a mapping $f(y)=(x_1,x_2)$.
> > > However, if there exist other values $(x_3, x_4)$ such that $x_3-x_4=y$, the mapping $f$ cannot distinguish between $(x_1, x_2)$ and $(x_3, x_4)$. This arises because the transformation from global to local coordinates is injective, while the reverse—from local to global—is inherently multi-valued.If we attempt to approximate this multi-valued mapping with a single-valued mapping $f$, the result would only represent a statistical average, not a true inverse.
> > >
> > > - **Noise**: Suppose we have data $s_i$ and corresponding observations $o_i=s_i+e_i$, where $e_i$ is i.i.d. noise drawn from $Normal(o,\sigma)$. To train a mapping $f$, we might minimize the mean squared error:
> > > $min_f \sum_i[f(s_i+e_i)-s_i]^2$.
> > > However, the minimizer in this case turns out to be the identity map, $f(o_i)=o_i$, meaning it simply outputs the noisy observation as-is. This implies that recovering a noiseless signal from noisy observations is fundamentally infeasible under these conditions.

---

> > > > ### Comment · Reviewer_ReJ3 · 2024-11-26
> > > >
> > > > Thank you for engaging with and responding to my questions.
> > > >
> > > > For the noise example, this MSE loss is not what I was suggesting. I was suggesting, for example, an NN where for an input noisy $o_i$, the loss is computed between the predicted $\pmb{s_i}$ and actual $s_i$ e.g. $\sum_i(\pmb{s_i}-s_i)^2$. This should be able to learn what I am suggesting.
> > > >
> > > > I take your point for the global coordinates example, but I am struggling to understand intuitively how your approach solves this problem. Using the the TigerDoor-alt example and following the 4 steps for GPO listed under Section 3.1, my understanding is that it would go:
> > > > 1. Collect trajectories in environment with $\mu$.
> > > > 2. Update the guider policy. Assuming perfect learning, the resulting policy would be $\left[\mu(a_L|s_L)=1, \mu(a_R|s_L)=0, \mu(a_R|s_R)=1, \mu(a_L|s_R)=0\right]$.
> > > > 3. Update the learner policy by minimising distance to $\mu$. Given an equal chance of the tiger being behind the left or right door, the learner policy should learn $\left[\pi(a_L|\emptyset)=0.5, \pi(a_R|\emptyset)=0.5\right]$, where $\emptyset$ shows the learner does not observe the tiger behind the door (or anything else, from my understanding).
> > > > 4. Backtracking step. Set $\left[\mu(a_L|s_L)=0.5, \mu(a_R|s_L)=0.5, \mu(a_R|s_R)=0.5, \mu(a_L|s_R)=0.5\right]$.
> > > > 5. Repeat.
> > > >
> > > > I cannot intuitively grasp how repeating steps 1-4 could result in the learner learning $\left[\pi(a_L|\emptyset)=1, \pi(a_R|\emptyset)=0\right]$, as demonstrated in your Figure 1. Essentially, I don't currently understand how larger rewards for a guide action result in higher probability of the learner taking that action. If you can point out my misunderstanding, I would appreciate it. The wording implies the backtracking step is what solves the issue, but I think explaining how it achieves that - in reference to one of the didactic examples - would strengthen the paper, as my intuition says this would run into the same problem as your global coordinates example. I would be happy to raise my score, if such an explanation were added.

---

> > > > > ### Author Response · Authors · 2024-11-26
> > > > >
> > > > > Thank you for your reply and thoughtful questions.
> > > > >
> > > > > Noise Example
> > > > > ---
> > > > > As the reviewer understands the global coordinates example, we’d like to further explain how the noise issue reflects a similar challenge involving multi-valued versus single-valued mappings. In this case, we aim to find a mapping $f(o_i)= \pmb{s_i}$, where $o_i=s_i+e_i$ and $e_i$ is the noise. The objective function you proposed $\sum_i(\pmb{s_i}-s_i)^2=\sum_i(f(s_i+e_i)-s_i)^2$ , faces a similar limitation.
> > > > >
> > > > > To see why, consider a noisy observation $o_1$. It’s possible for there to exist multiple corresponding ground truths, $s_1$ and $s_2$, such that: $o_1=s_1+e_1=s_2+e_2$, where $e_1$ and $e_2$ are drawn from a normal distribution. In this case, $f(o_1)$ cannot determine whether $s_1$ or $s_2$ is the correct underlying signal.
> > > > >
> > > > > TigerDoor-Alt Problem
> > > > > ---
> > > > > We appreciate your engagement with the core of our method. To clarify how the iterative process works and address your intuition:
> > > > >
> > > > > Initially, the guider’s policy is uniform:
> > > > >  $\mu(\cdot|s_L)=(0.5,0.5), \mu(\cdot|s_R)=(0.5,0.5)$.
> > > > >
> > > > > **Step-by-Step Breakdown**:
> > > > > - **Guider Policy Update**:
> > > > > After an update step, the guider’s policy shifts to reflect the reward structure. For instance:
> > > > > $\mu(\cdot|s_L)=(0.7,0.3), \mu(\cdot|s_R)=(0.4,0.6)$.
> > > > > The key here is that the higher reward for $(s_L,a_L)$ results in a larger gradient update compared to $(s_R,a_R)$, biasing $\mu(\cdot|s_L)$ more strongly toward $a_L$.
> > > > > - **Learner Policy Update**:
> > > > > The learner imitates the guider, resulting in:
> > > > > $\pi=((0.7+0.4)/2,(0.3+0.6)/2)=(0.55,0.45)$.
> > > > > This adjustment brings the learner’s policy closer to the optimal policy
> > > > > - **Guider Backtracking**:
> > > > > After backtracking, the guider’s policy is reset to match the learner:
> > > > > $\mu(\cdot|s_L)= \mu(\cdot|s_R)=(0.55,0.45)$.
> > > > > - **Subsequent Iteration**:
> > > > > In subsequent iterations, this process continues with initial policy $(0.55,0.45)$, and result in the learner’s policy gradually improving. For example, in the next iteration, we will observe:
> > > > > $\pi(a_L)>0.55$ and $\pi(a_R)<0.45$.
> > > > > This iterative refinement drives the learner toward the optimal policy.
> > > > >
> > > > > The critical factor is that higher rewards for specific guider actions result in larger updates, which the learner captures through imitation. Simultaneously, the backtracking step ensures that the guider remains aligned with the learner, fostering consistent improvement.
> > > > >
> > > > > This dynamic is also observable in Figure 1(b). Initially, the PPO+BC curve approaches the optimal reward, reflecting that the average of guider’s policy tends to favor $a_L$ early on. However, as the iteration progresses, the curve declines to the level of a random policy, indicating that the guider’s policy eventually converges, resulting an average between $a_L$ and $a_R$.

---

> > > > > > ### Comment · Reviewer_ReJ3 · 2024-11-27
> > > > > >
> > > > > > Thank you very much for this explanation. I appreciate your time and effort. I think particularly the last sentence very much clarified the method, and the effect of the backtracking. If you could fit the above explanation into even an appendix, I think it would be helpful for readers less familiar with similar methods. I will raise my score.

---

> > > > > > > ### Author Response · Authors · 2024-11-27
> > > > > > >
> > > > > > > Thank you for your thoughtful feedback and for taking the time to engage deeply with our work. We’re glad the explanation clarified the method and have added it to the appendix.

---

### Official Review · Reviewer_KrL3 · 2024-11-03

**Soundness:** 2
**Presentation:** 3
**Contribution:** 2
**Rating:** 1
**Confidence:** 5

**Summary:**

This paper introduces Guided Policy Optimization (GPO), a framework that co-trains a guider and a learner in partially observable reinforcement learning (RL) environments. The guider utilizes supplementary information to enhance training while maintaining alignment with the learner's policy. The guider is trained with RL, and the learner primarily learns through Imitation Learning (IL). This approach achieves optimality comparable to direct RL. Empirical evaluations demonstrate strong performance across various tasks, including continuous control with partial observability, noise, and memory challenges, significantly outperforming baseline methods.

**Strengths:**

- The paper addresses a pertinent challenge by leveraging additional information to improve training in RL environments.-
- It is well-structured, clearly written, and easy to follow.
- Empirical studies show consistent improvements over baseline approaches across multiple benchmark domains.

**Weaknesses:**

- The application of IL here is unclear, as there are no demonstrations, making it resemble a model distillation approach for partially observable settings. Restructuring the paper to emphasize knowledge distillation over imitation learning might better reflect the method's contributions.
- Several modifications and "tricks" are presented, but the paper lacks adequate empirical or theoretical justification for each.
- There are no comparisons with state-of-the-art baselines, which limits the ability to gauge the method’s relative effectiveness.
- There is no discussion of related work.

**Questions:**

Have you tested the approach on real-world robots trained with data from simulations?

---

> ### Author Response · Authors · 2024-11-18
>
> Thanks for your review.
>
> 1.**The application of IL here is unclear, as there are no demonstrations, making it resemble a model distillation approach for partially observable settings. Restructuring the paper to emphasize knowledge distillation over imitation learning might better reflect the method's contributions**.
>
> Thank you for the suggestion. However, our paper does not primarily focus on knowledge distillation. Knowledge distillation approaches generally focus on improving the student's learning process. In contrast, our method is centered on training a better teacher. The techniques proposed in our paper are specifically designed to enhance the teacher's performance, enabling it to provide more effective supervision to the student.
>
> 2.**Several modifications and "tricks" are presented, but the paper lacks adequate empirical or theoretical justification for each**.
>
> We disagree with this point. The components of our method are thoroughly justified, as summarized below:
>
> | Modifications/Tricks | Theoretical Justification | Empirical Justification |
> | :--- | :--- | :--- |
> |GPO style update| Proposition 1: Improvement and optimality guarantee | Section 4.1: Performance validation on optimality |
> |RL auxiliary loss for GPO-penalty| Proposition 2: Validity of the RL loss| Sections 4.1 and 4.2: Empirical results on its effectiveness|
> | Additional clip and mask for GPO-penalty| Not required, as the approach is straightforward and intuitive | Sections 4.2 and 4.3: Performance gains and Figure 5: More flexible KL control|
>
> We believe all components of our method are well-supported by either theoretical proofs or empirical evidence. If specific points are unclear or missing, we would appreciate it if you could **explicitly** highlight which aspects require further justification.
>
>
> 3.**There are no comparisons with state-of-the-art baselines, which limits the ability to gauge the method’s relative effectiveness**.
>
> Could you please **specify** which baselines you believe are missing?
>
> 4.**There is no discussion of related work**.
>
> It is provided in Appendix A.
>
> 5.**Have you tested the approach on real-world robots trained with data from simulations**?
>
> No.

---

> > ### Comment · Reviewer_KrL3 · 2024-12-02
> >
> > Thank you for your detailed response and clarifications. However, I disagree with your assertion that your method is fundamentally distinct from policy distillation. While you emphasize improving the teacher (guider), the ultimate purpose of this improvement is still to enhance the student’s learning, which aligns closely with the objectives of policy distillation: effective transfer of knowledge between policies. The distinction you draw does not sufficiently separate your approach from the broader framework of policy distillation techniques.
> >
> > Additionally, I find it problematic that your approach is framed within the context of imitation learning (IL) algorithms, as it does not rely on demonstrations—a key assumption of IL. To provide a more robust evaluation of your method, I suggest including comparisons to state-of-the-art baselines such as original ADVISOR, TGRL, and SAC.

---

> > > ### Author Response · Authors · 2024-12-03
> > >
> > > Thank you for your feedback.
> > >
> > > Reviewer KrL3 suggested that our method aligns with policy distillation, while Reviewer pw6H identified it as teacher-student learning. In our paper, we present the broader context of imitation learning, which encompasses both frameworks. However, we deliberately do not categorize our work strictly as policy distillation, teacher-student learning, or imitation learning because our method introduces a novel framework. The closest related approach is Guided Policy Search, which is why we refer to our method as Guided Policy Optimization.
> > >
> > > It’s worth noting that Guided Policy Search itself could arguably be framed as policy distillation or teacher-student learning, yet it does not explicitly label itself as such. We believe that framing our method under any specific category—be it policy distillation, teacher-student learning, or otherwise—can be a helpful perspective but should not be considered a fundamental weakness of the paper. In addition, the state-of-the-art baselines you mentioned, such as ADVISOR and TGRL, are not policy distillation methods either.
> > >
> > > Regarding your suggestion to include comparisons with ADVISOR and TGRL, we kindly refer you to the general rebuttal above and Appendix E, where we explain why these methods are not competitive to our method.

---

### Official Review · Reviewer_pw6H · 2024-11-04

**Soundness:** 3
**Presentation:** 3
**Contribution:** 1
**Rating:** 3
**Confidence:** 4

**Summary:**

The author propose a teacher-student RL framework that solves POMDP where the teacher takes privileged information and the student performs imitation learning.

**Strengths:**

The problem class is important and hard to solve for traditional methods. POMDPs are known to be hard to learn. The proposed idea is interesting and the results are satisfactory.
In particular, the backtracking step seems the most interesting part, as other section has already been discussed in the relevant works (see below).

**Weaknesses:**

I found the proposed work largely overlap with the existing work [1], which is public since Feb 2024. [1] proposed a method for solving POMDP, where the teacher takes in state of the underlaying MDP and perform policy mirror descent, and the student performs imitation learning. In [1], the teacher collects the data through environmental interaction, and the student perform offline imitation, which is exactly the same procedure described in this work. The additional loss term L4 seems to be exactly the asymmetric loss in [1] if written out explicitly in the form of advantage function times a ratio. A later work [2] also proposed the same framework while utilizing PPO as backbone.

[1]. Wu, Feiyang, et al. "Learn to Teach: Improve Sample Efficiency in Teacher-student Learning for Sim-to-Real Transfer." arXiv preprint arXiv:2402.06783 (2024).

[2] Wang, Hongxi, et al. "CTS: Concurrent Teacher-Student Reinforcement Learning for Legged Locomotion." IEEE Robotics and Automation Letters (2024).

In view of this, the real contribution of this work seems minimal. A specific implementation using PPO seems more of a computational trick for stable training. This leave the real novelty with the back tracking step, i.e., setting $\mu^k = \pi^k$ at each iteration. But I am not sure how much novelty this holds.

Despite this, I have a few additional concern:
1. Due to the backtracking step, prop 1 and 2 seems to hold little value, assuming one can minimize $KL(\mu||\pi)$ accurately enough.
2. The overall usage of notation are abusive. The paper writing itself reflects its confusion. For example, in eq 9 and 10, $o_g$ (supposed to given to guider $\mu$) and $o_l$ seem to be out of place.
3. In the experiments section, it's rather strange to see PPO+BC can perform worse than PPO itself. Additionally PPO-V (asymmetric ppo) also performs reasonably well and in most cases achieves on par performance as the best in class. Then it makes one wonder what's really contributing to the performance of the proposed method. I think the only interesting case is rather GPO without the backtracking. However, I do not see the experiments on such case.
4. One can simply train a standard teacher student learning paradigm where we obtain a teacher first and then train a student. This is the standard idea dealing with POMDPs in robot learning field. This is crucial for comparison since if the proposed method cannot achieve the same level of performance, then it holds little practical value for the trained learner's policy. However, I do not see this set of experiments.

**Questions:**

The reward function and the ratio are both denoted as $r$. It is very hard to discern through the texts.

**Details Of Ethics Concerns:**

Due to fact that this work has high correlation with the prior work of [1] and [2], I have to express my concern of originality of this paper. It is one thing to propose a new idea or make an extension, but another to avoid cite prior works and make seemingly deliberate alternation of the context so as to avoid suspicion. I am under the impression that in robotics learning community, teacher-student learning for POMDPs are already popular and well known, and the paper itself cites numerous relevant papers e.g. (Chen et al,. 2020). But the authors chose to avoid using teacher-student, or privileged learning which are standard description for the approach. This makes wonder if this is intentional as to avoid association with the proposed teacher student learning field so that the authors would be asked to compare to [1] or [2].

Additionally, it is rather less known in robotics learning community that Policy Mirror Descent, the utilization of which is also an emphasis in [1]. This adds my concern on similarity of the ideas these two papers have.

In any case, I'd hope this is a coincident, or an honest mistake.

---

> ### Author Response · Authors · 2024-11-14
> **Response to Ethics Concerns Raised by Reviewer pw6H**
>
> This part only focuses on **ethical issues**.
>
> **General Clarification**:
> -
> There are two distinct lines of work addressing the problem in question: **Teacher-Student Learning (TSL)** and **Guided Policy Search (GPS)**.
>
> 1. **TSL**: This typically assumes a predefined or trained teacher and focuses on optimizing the student’s learning process.
>
> 2. **GPS**: This simultaneously optimizes both teacher and student policies while **enforcing alignment**. A key characteristic of the GPS framework is that it constrains the teacher's policy to remain close to the student's, which is not typically the case in TSL.
>
> The methods presented in [1] and [2], as referred to by the reviewer, align with TSL, whereas our approach—featuring a **backtracking** step—falls under the GPS framework. We have explicitly mentioned this in our paper (line 50 and line 137).
>
> **Specific Comparisons:**
> -
> To make this distinction more explicit, we provide a detailed comparison between our approach and the framework in [1] (assuming it uses PPO). Below is a summary of key differences across various methods:
>
> | Algorithm| Train $\mu$ | Behavioral policy | Train $\pi$ | Value function | backtrack $\mu$|
> | :---: | :---: | :---: |:---: | :---: | :---: |
> | PPO    | - | $\pi(a\|o_l)$ | PPO | $V(o_l)$ | - |
> | PPO+V | - | $\pi(a\|o_l)$ | PPO | $V(o_g)$ | - |
> | PPO+BC | PPO | $\mu(a\|o_g)$ | BC | $V(o_g)$ | No |
> | **L2T-RL**[1] | PPO | $\mu(a\|o_g)$ | BC+PPO | $V(o_g)$ | No |
> | A2D | PPO | $\pi(a\|o_l)$ | BC | $V(o_l)$ | No |
> | ADVISOR-co | PPO | $\pi(a\|o_l)$ | BC+PPO | $V(o_l)$ | No |
> | GPO-naive | PPO | $\mu(a\|o_g)$ | BC | $V(o_g)$ | Yes |
> | GPO-clip/penalty | PPO | $\mu(a\|o_g)$ | BC+PPO | $V(o_g)$ | Yes |
> | GPS | trajectory optimization | teacher $\mu$ | BC | - | Yes |
>
> As shown, without the **backtracking** step, these methods are all fundamentally similar, differing only in their specific combinations of behavioral policies and learner training schedules.
> Our method, however, introduces a **backtracking** step, which places it firmly within the **GPS framework**. In essence, GPO-naive can be regard as direct implementation of GPS using PPO. Our focus is on how to make this idea theoretically sound and practically viable, not only by incorporating several "computational tricks for stable training," as noted by the reviewer.
>
> Regarding [1], it is important to note that it does not provide any improvement or convergence guarantees. Our experiments (to be provided later) demonstrate that the method in [1] **fails to converge**, much like the **PPO+BC** framework. The underlying issue, as explained in point 4 of Section 4.2, is that if the teacher’s policy is used as the behavioral policy **without backtracking**, the student may not learn anything from BC. Consequently, the RL training for the student becomes offline, necessitating offline RL algorithms to avoid divergence. This issue is evident in Figure 2 of [1] itself, where the student diverges in two tasks, further supporting our claims.
>
> The framework in [2] proposes a mixed training approach for teacher and student using PPO. This approach is fundamentally different from [1] and from all methods compared in our paper.
>
> As for the term **Policy Mirror Descent**, it is a well-known concept in the RL community. For instance, we cite “William H. Montgomery and Sergey Levine. Guided policy search via approximate **mirror descent**” as an example of its usage within the GPS framework. Its usage in our work is consistent with established norms in RL research and not unique to [1].
>
> We hope this response adequately addresses the **ethical concerns** raised by Reviewer pw6H. If there are additional questions or points for clarification specifically regarding the ethical concerns (excluding other aspects), we kindly request further feedback to ensure all concerns are resolved.

---

> > ### Comment · Reviewer_pw6H · 2024-11-18
> > **Thank you for the reponse**
> >
> > I thank the authors for the clarification. While it looks like this is a coincidence, there are still some general concern not being addressed properly:
> > 1. Teacher-student learning is clearly not just student learning, in which case people refer as Imitation Learning, or Behavior Cloning.
> > 2. The convergence of L2T-RL seems to be demonstrated in the paper.
> > 3. [2] can be seen as a variation of the proposed method as performing backtracking periodically.
> > 4. Mirror descent, originally as a first order optimization method, has not been applied in RL until recent development in both theoretical and practical fields. The term Policy Mirror Descent (which applies mirror descent type update to policy improvement steps) carries different meanings between general optimization and RL.

---

> > > ### Author Response · Authors · 2024-11-18
> > > **Response to Ethics Concerns**
> > >
> > > 1. **Teacher-student learning is clearly not just student learning**.
> > >
> > > Is there a principled or formal definition of teacher-student learning (TSL) in RL?
> > >
> > > To our understanding, TSL originates from supervised learning. While some training may occur for the teacher, the primary focus is on knowledge distillation to the student. The recent TSL RL algorithm [3] we referenced also defines TSL as being agnostic to the teacher.
> > > If the reviewer has a different perspective or additional insights, we would greatly appreciate further clarification or relevant references.
> > >
> > > 2. **The convergence of L2T-RL seems to be demonstrated in the paper**.
> > >
> > > We could not find theoretical results supporting the convergence of L2T-RL [1]. Are we overlooking something?
> > >
> > > Figure 11 in Appendix F of our paper includes the results of L2T-RL, showing that it **fails similarly to PPO+BC**.
> > > A simple reasoning illustrates why both PPO+BC and L2T-RL fail: consider the **TighterDoor** task. If the behavioral policy serves as the teacher but is **not aligned with the student** due to certain constraints, the teacher’s optimal policy will not include the "listen" action, which is crucial for the student. Initially, the student might show improvement because the teacher randomly selects the "listen" action. However, as the teacher converges, training data for the student no longer contains this action, leading to a significant **decline** in the student’s performance.
> > >
> > > This behavior is evident from the performance drop of PPO+BC and L2T-RL in Figure 11. Additionally, the _Hopper_ and _Walker2d_ results in Figure 2 of L2T-RL’s paper appear to support this observation.
> > >
> > > 3. **[2] can be seen as a variation of the proposed method, performing backtracking periodically**.
> > >
> > > This description is vague. Is [2] **theoretically equivalent** to our work, or does it merely share a **similar idea**?
> > > Since [2] does not include theoretical results, we cannot deduce the extent of its similarity to our method. Furthermore, it is unclear for us how backtracking is performed periodically.
> > >
> > > To our understanding, [2] trains the student encoder to approximate the teacher’s encoder, while the teacher’s encoder remains unaffected by the student. As such, we do not observe evidence of backtracking for the teacher.
> > >
> > > 4. **About policy mirror descent**.
> > >
> > > If the reviewer examines [4] closely, the following excerpt from it aligns with the problem our paper addresses and motivated us to leverage the properties of mirror descent to prove Proposition 1:
> > > >Mirror descent solves this optimization by alternating between two steps at each iteration $k$:
> > > >$p^k \gets \arg\min_p E_{p(\tau)}\big[\sum_{t=1}^T l(x_t, u_t)\big] \text{ s.t. } D(p, \pi^k) \leq \epsilon, \pi^{(k+1)} \gets \arg\min_{\pi \in \Pi} D(p^k, \pi)$.
> > >
> > > where $p$ is teacher, $\pi$ is student and $l$ is cost.
> > >
> > > Directly proving Proposition 1 for standard RL would be more complex, which is why we adopted this approach.
> > > Additionally, incorporating policy mirror descent into [1] appears unnecessary. Replacing it with PPO or SAC achieves equivalent results, as [1] does not utilize any theoretical properties of policy mirror descent.
> > >
> > > [1] Wu, Feiyang, et al. "Learn to Teach: Improve Sample Efficiency in Teacher-student Learning for Sim-to-Real Transfer." arXiv preprint arXiv:2402.06783 (2024).
> > >
> > > [2] Wang, Hongxi, et al. "CTS: Concurrent Teacher-Student Reinforcement Learning for Legged Locomotion." IEEE Robotics and Automation Letters (2024).
> > >
> > > [3] Shenfeld, Idan, et al. "Tgrl: An algorithm for teacher guided reinforcement learning." International Conference on Machine Learning. PMLR, 2023.
> > >
> > > [4] Montgomery W, Levine S. Guided policy search as approximate mirror descent. arXiv preprint arXiv:1607.04614, 2016.

---

> ### Author Response · Authors · 2024-11-18
> **Response to other concerns**
>
> We appreciate the reviewer’s time and feedback. However, according to the reviews review, we believe there may have been a misunderstanding of the **core idea** behind our method. This may stem from our lack of a detailed introduction to Guided Policy Search in the original manuscript. To address this, we recommend the reviewer refer to the general rebuttal, where we provide a concise summary of GPS and its critical role in our approach.
>
> The **backtracking** step, in particular, is fundamental to our method, as it ensures **theoretical correctness** and enables effective **utilization of teacher guidance** with privileged information. Below, we address the specific concerns raised in the review. Some points may overlap with earlier responses.
>
> 1. **Due to the backtracking step, prop 1 and 2 seems to hold little value, assuming one can minimize $KL(\mu||\pi)$ accurately enough**.
>
> Achieving the theoretical results is **nontrivial** because we can only minimize this KL from one side.
> Specifically, we can accurately minimize $\min_\mu KL(\mu||\pi)$ but cannot minimize $\min_\pi KL(\mu||\pi)$ small enough due to limited observation.
> This makes the policy improvement of $\pi$ crucial. Without it, it is unclear whether the GPO update merely adjusts $\mu$ through RL and pulls it back through KL, leaving $\pi$ unchanged—or worse, degrading its performance. Our theoretical results provide guarantees for the correctness of these updates, which cannot be assumed otherwise.
>
> 2. **In eq 9 and 10, $o_g$ and $o_l$ seem to be out of place**.
>
> Thank you for pointing this out. This is indeed a typo, and we have corrected it in the revised version of the paper.
>
> 3. **Why PPO+BC perform worse than PPO**.
>
> As discussed in Section 4.2, this occurs because PPO+BC uses the guider’s policy to collect experiences **without incorporating a backtracking step**. This makes the teacher’s policy “impossibly good” for the student, causing the BC-loss to hinder training.
> A similar issue is observed with a pretrained teacher, as shown in Figure 9 of the Appendix E. PPO+BC underperforms PPO due to the teacher's privileged access to information. In fact, many prior works address this situation by switching the student’s training to pure RL, as discussed in the general rebuttal.
>
> 4. **PPO-V (asymmetric ppo) also performs reasonably well and in most cases achieves on par performance as the best in class**.
>
> While asymmetric PPO performs better than other baselines, our methods (GPO-clip and GPO-penalty) outperform it in **almost all** tasks in MuJoCo. Furthermore, in the POPGym benchmarks, asymmetric PPO performs similarly to PPO, whereas our methods **significantly** outperform it.
>
> 5. **I think the only interesting case is rather GPO without the backtracking. However, I do not see the experiments on such case**.
>
> The necessity of the backtracking step for GPO has been clarified in the responses above. Additionally, GPO-naive without backtracking is essentially equivalent to PPO+BC. This demonstrates why the backtracking step is indispensable for achieving the theoretical and practical benefits of our method.
>
> 6. **Obtain a teacher first and then train a student**.
>
> Thank you for this suggestion. We conducted experiments based on this idea. Please refer to the general rebuttal and Appendix E for detailed results and discussion.

---

### Author Response · Authors · 2024-11-18
**General Concerns Raised by Reviewers**

1. **Why not first train a teacher based on full observation and then apply it to train a student?**
---
We address this by emphasizing that even **without considering the additional time and data** required to train a good teacher, this approach **does not work** in practice. Training a teacher first and then applying IL methods will ultimately cause SOTA methods, such as TGRL or ADVISOR, to **degenerate into pure RL**.

Intuitively, consider the **TigerDoor** task: a trained teacher will fail to provide any meaningful supervision to the student, since it will never choose the "listen" action. Teacher-student methods struggle to mimic the teacher and inevitably revert to pure RL.
By contrast, our method, GPO-naive, achieves optimal performance **without requiring RL training** for the learner. This demonstrates that our approach effectively utilizes teacher guidance.

To provide further evidence, we present experimental results (see Appendix E) that compare PPO+BC, ADVISOR, and TGRL in scenarios where a trained teacher is available:

1.**Unprivileged Teacher**: When both the teacher and the student operate under the same partial observation setting, teacher supervision significantly improves sample efficiency.

2.**Privileged Teacher**: When the teacher is trained with full observation but the student operates in a partially observable setting, teacher supervision becomes ineffective. In this case, ADVISOR and TGRL degrade into PPO and SAC, respectively. Additionally, PPO+BC underperforms even pure PPO due to the detrimental effects of BC during training.

While our setup may seem uncommon, it reflects a general and practical scenario in real-world applications: starting with additional information in a POMDP and determining how to use it effectively. Our findings reveal that directly training a teacher often proves futile.
For example, as our experiments suggest, a teacher trained with velocity information cannot supervise a student without access to velocity; a teacher trained in noise-free settings cannot guide a student facing noisy observations.
Prior works focus predominantly on **student training**, ensuring that algorithms switch to pure RL if teacher supervision is ineffective. In contrast, our paper emphasizes **teacher training**, ensuring it provides actionable supervision to the student.

2. **Related Works**.
---
We acknowledge the need to include an introduction to the GPS family of methods, as some reviewers noted this omission made our motivations and framework less clear. Below, we summarize GPS and its relevance to our work.

Unlike direct policy search methods (e.g., REINFORCE), GPS does not optimize policy parameters directly. Instead, it introduces an intermediate agent and employs trajectory optimization to learn a time-varying linear-Gaussian policy, which is then used to train a neural network policy through supervised learning. The GPS procedure has two key phases:
- **Control Phase**: A control policy interacts with the environment to minimize costs while ensuring learnability by the neural network policy.
- **Supervised Phase**. The neural network policy is trained to mimic the control policy similar to Behavioral Cloning.

Among the foundational GPS works [1][2][3], our framework most closely aligns with [3], which incorporates a KL constraint to ensure consistency between the control policy and the neural network policy. Over the years, GPS has been extended in various directions, such as: handling unknown dynamics [4]; simplifying optimization via mirror descent [5]; integrating with path integral methods [6] and LQR [7]; Incorporating memory models [8].

Our GPO framework inherits the core GPS idea of **training a guiding policy to supervise the learner policy while maintaining alignment** between them.

[1] Levine S, Koltun V. Guided policy search. International Conference on Machine Learning. 2013.

[2] Levine S, Koltun V. Variational policy search via trajectory optimization. Advances in neural information processing systems. 2013.

[3] Levine S, Koltun V. Learning complex neural network policies with trajectory optimization. International Conference on Machine Learning. 2014.

[4] Levine S, Abbeel P. Learning neural network policies with guided policy search under unknown dynamics. Advances in Neural Information Processing Systems. 2014.

[5] Montgomery W, Levine S. Guided policy search as approximate mirror descent. arXiv preprint arXiv:1607.04614, 2016.

[6] Chebotar Y, Kalakrishnan M, Yahya A, et al. Path integral guided policy search. 2017 IEEE international conference on robotics and automation. IEEE, 2017.

[7] Chebotar Y, Hausman K, Zhang M, et al. Combining model-based and model-free updates for trajectory-centric reinforcement learning[J]. arXiv preprint arXiv:1703.03078, 2017.

[8] Zhang, Marvin, et al. "Learning deep neural network policies with continuous memory states." 2016 IEEE international conference on robotics and automation. IEEE, 2016.

---

### Meta-Review · Area_Chair_rKkY · 2024-12-19

**Metareview:**

The paper studied the POMDP setting where a teacher has access to full information and the student only has access to the partial observations. It proposed a framework which co-traines both the teacher and the student. The paper argue that this co-training is important because it allows the student to better mimic (imitate) the teacher during learning.

Weaknesses identified by the reviewers include the limited novelty of the proposed approach when compared to prior work and insufficient experiments, particularly the comparison to some state-of-the-art approaches.

**Additional Comments On Reviewer Discussion:**

While the reviewers acknowledged the authors' effort in rebuttal, they were unconvinced after the rebuttal phase. The reviewers were unconvinced that the standard teacher-student-based baselines are less effective than the proposed approach. While the authors provided some examples, there were no formal theoretical statements to support them. This leads one of the reviewers to argue strongly for a direct comparison to one prior work. Since the proposed approach falls into the same setting as prior teacher-student-based work, and if the authors want to claim their approach is more effective, we encourage them to compare their approach to prior approaches explicitly.

---

### Decision · Program_Chairs · 2025-01-22

Reject